# On Structured Prediction Theory with Calibrated Convex Surrogate Losses

**Anton Osokin**
INRIA/ENS,* Paris, France
HSE,† Moscow, Russia

**Francis Bach**
INRIA/ENS,* Paris, France

**Simon Lacoste-Julien**
MILA and DIRO
Université de Montréal, Canada

## Abstract

We provide novel theoretical insights on structured prediction in the context of *efficient* convex surrogate loss minimization with consistency guarantees. For any task loss, we construct a convex surrogate that can be optimized via stochastic gradient descent and we prove tight bounds on the so-called "calibration function" relating the excess surrogate risk to the actual risk. In contrast to prior related work, we carefully monitor the effect of the exponential number of classes in the learning guarantees as well as on the optimization complexity. As an interesting consequence, we formalize the intuition that some task losses make learning harder than others, and that the classical 0-1 loss is ill-suited for structured prediction.

## 1 Introduction

Structured prediction is a subfield of machine learning aiming at making multiple interrelated predictions simultaneously. The desired outputs (labels) are typically organized in some structured object such as a sequence, a graph, an image, etc. Tasks of this type appear in many practical domains such as computer vision [34], natural language processing [42] and bioinformatics [19].

The structured prediction setup has at least two typical properties differentiating it from the classical binary classification problems extensively studied in learning theory:
1. Exponential number of classes: this brings both additional computational and statistical challenges. By *exponential*, we mean exponentially large in the size of the natural dimension of output, e.g., the number of all possible sequences is exponential w.r.t. the sequence length.
2. Cost-sensitive learning: in typical applications, prediction mistakes are *not* all equally costly. The prediction error is usually measured with a highly-structured task-specific loss function, e.g., Hamming distance between sequences of multi-label variables or mean average precision for ranking.

Despite many algorithmic advances to tackle structured prediction problems [4, 35], there have been relatively few papers devoted to its theoretical understanding. Notable recent exceptions that made significant progress include Cortes et al. [13] and London et al. [28] (see references therein) which proposed data-dependent generalization error bounds in terms of popular empirical convex surrogate losses such as the structured hinge loss [44, 45, 47]. A question not addressed by these works is whether their algorithms are *consistent*: does minimizing their convex bounds with infinite data lead to the minimization of the task loss as well? Alternatively, the structured probit and ramp losses are consistent [31, 30], but non-convex and thus it is hard to obtain computational guarantees for them. In this paper, we aim at getting the property of consistency for surrogate losses that can be *efficiently* minimized with guarantees, and thus we consider *convex* surrogate losses.

The consistency of convex surrogates is well understood in the case of binary classification [50, 5, 43] and there is significant progress in the case of multi-class 0-1 loss [49, 46] and general multi-

class loss functions [3, 39, 48]. A large body of work specifically focuses on the related tasks of ranking [18, 9, 40] and ordinal regression [37].

**Contributions.** In this paper, we study consistent convex surrogate losses specifically in the context of an exponential number of classes. We argue that even while being consistent, a convex surrogate might not allow efficient learning. As a concrete example, Ciliberto et al. [10] recently proposed a consistent approach to structured prediction, but the constant in their generalization error bound can be exponentially large as we explain in Section 5. There are two possible sources of difficulties from the optimization perspective: to reach adequate accuracy on the *task* loss, one might need to optimize a surrogate loss to exponentially small accuracy; or to reach adequate accuracy on the *surrogate* loss, one might need an exponential number of algorithm steps because of exponentially large constants in the convergence rate. We propose a theoretical framework that jointly tackles these two aspects and allows to judge the feasibility of efficient learning. In particular, we construct a *calibration function* [43], i.e., a function setting the relationship between accuracy on the surrogate and task losses, and normalize it by the means of convergence rate of an optimization algorithm.

Aiming for the simplest possible application of our framework, we propose a family of convex surrogates that are consistent for any given task loss and can be optimized using stochastic gradient descent. For a special case of our family (quadratic surrogate), we provide a complete analysis including general lower and upper bounds on the calibration function for any task loss, with exact values for the 0-1, block 0-1 and Hamming losses. We observe that to have a tractable learning algorithm, one needs both a structured loss (not the 0-1 loss) and appropriate constraints on the predictor, e.g., in the form of linear constraints for the score vector functions. Our framework also indicates that in some cases it might be beneficial to use non-consistent surrogates. In particular, a non-consistent surrogate might allow optimization only up to specific accuracy, but exponentially faster than a consistent one.

We introduce the structured prediction setting suitable for studying consistency in Sections 2 and 3. We analyze the calibration function for the quadratic surrogate loss in Section 4. We review the related works in Section 5 and conclude in Section 6.

## 2 Structured prediction setup

In structured prediction, the goal is to predict a structured output $y \in \mathcal{Y}$ (such as a sequence, a graph, an image) given an input $x \in \mathcal{X}$. The quality of prediction is measured by a task-dependent *loss function* $L(\hat{y}, y \mid x) \geq 0$ specifying the cost for predicting $\hat{y}$ when the correct output is $y$. In this paper, we consider the case when the number of possible predictions and the number of possible labels are both finite. For simplicity,[1] we also assume that the sets of possible predictions and correct outputs always coincide and do not depend on $x$. We refer to this set as the set of labels $\mathcal{Y}$, denote its cardinality by $k$, and map its elements to $1, \ldots, k$. In this setting, assuming that the loss function depends only on $\hat{y}$ and $y$, but not on $x$ directly, the loss is defined by a loss matrix $L \in \mathbb{R}^{k \times k}$. We assume that all the elements of the matrix $L$ are non-negative and will use $L_{\max}$ to denote the maximal element. Compared to multi-class classification, $k$ is typically exponentially large in the size of the natural dimension of $y$, e.g., contains all possible sequences of symbols from a finite alphabet.

Following standard practices in structured prediction [12, 44], we define the prediction model by a *score function* $\mathfrak{f} : \mathcal{X} \to \mathbb{R}^k$ specifying a score $\mathfrak{f}_y(x)$ for each possible output $y \in \mathcal{Y}$. The final prediction is done by selecting a label with the maximal value of the score

$$\text{pred}(\mathfrak{f}(x)) := \operatorname*{argmax}_{\hat{y} \in \mathcal{Y}} \mathfrak{f}_{\hat{y}}(x), \tag{1}$$

with some fixed strategy to resolve ties. To simplify the analysis, we assume that among the labels with maximal scores, the predictor always picks the one with the smallest index.

The goal of prediction-based machine learning consists in finding a predictor that works well on the unseen test set, i.e., data points coming from the same distribution $\mathcal{D}$ as the one generating the training data. One way to formalize this is to minimize the generalization error, often referred to as the actual (or population) *risk* based on the loss $L$,

$$\mathcal{R}_L(\mathfrak{f}) := \mathbf{E}_{(x,y) \sim \mathcal{D}} \, L\big(\text{pred}(\mathfrak{f}(x)), y\big). \tag{2}$$

Minimizing the actual risk (2) is usually hard. The standard approach is to minimize a *surrogate risk*, which is a different objective easier to optimize, e.g., convex. We define a surrogate loss as a function

$\Phi : \mathbb{R}^k \times \mathcal{Y} \to \mathbb{R}$ depending on a score vector $\boldsymbol{f} = \mathfrak{f}(\boldsymbol{x}) \in \mathbb{R}^k$ and a target label $\boldsymbol{y} \in \mathcal{Y}$ as input arguments. We denote the $\boldsymbol{y}$-th component of $\boldsymbol{f}$ with $f_{\boldsymbol{y}}$. The surrogate risk (the $\Phi$-risk) is defined as

$$\mathcal{R}_\Phi(\mathfrak{f}) := \mathbb{E}_{(\boldsymbol{x},\boldsymbol{y}) \sim \mathcal{D}} \; \Phi(\mathfrak{f}(\boldsymbol{x}), \boldsymbol{y}), \tag{3}$$

where the expectation is taken w.r.t. the data-generating distribution $\mathcal{D}$. To make the minimization of (3) well-defined, we always assume that the surrogate loss $\Phi$ is bounded from below and continuous.

Examples of common surrogate losses include the structured hinge-loss [44, 47] $\Phi_{\text{SSVM}}(\boldsymbol{f}, \boldsymbol{y}) := \max_{\hat{\boldsymbol{y}} \in \mathcal{Y}} (f_{\hat{\boldsymbol{y}}} + L(\hat{\boldsymbol{y}}, \boldsymbol{y})) - f_{\boldsymbol{y}}$, the log loss (maximum likelihood learning) used, e.g., in conditional random fields [25], $\Phi_{\log}(\boldsymbol{f}, \boldsymbol{y}) := \log(\sum_{\hat{\boldsymbol{y}} \in \mathcal{Y}} \exp f_{\hat{\boldsymbol{y}}}) - f_{\boldsymbol{y}}$, and their hybrids [38, 21, 22, 41].

In terms of task losses, we consider the unstructured *0-1 loss* $L_{01}(\hat{\boldsymbol{y}}, \boldsymbol{y}) := [\hat{\boldsymbol{y}} \neq \boldsymbol{y}]$,[2] and the two following structured losses: *block 0-1 loss* with $b$ equal blocks of labels $L_{01,b}(\hat{\boldsymbol{y}}, \boldsymbol{y}) := [\hat{\boldsymbol{y}} \text{ and } \boldsymbol{y} \text{ are not in the same block}]$; and (normalized) *Hamming loss* between tuples of $T$ binary variables $y_t$: $L_{\text{Ham},T}(\hat{\boldsymbol{y}}, \boldsymbol{y}) := \frac{1}{T} \sum_{t=1}^{T} [\hat{y}_t \neq y_t]$. To illustrate some aspects of our analysis, we also look at the *mixed loss* $L_{01,b,\eta}$: a convex combination of the 0-1 and block 0-1 losses, defined as $L_{01,b,\eta} := \eta L_{01} + (1 - \eta) L_{01,b}$ for some $\eta \in [0, 1]$.

# 3 Consistency for structured prediction

## 3.1 Calibration function

We now formalize the connection between the actual risk $\mathcal{R}_L$ and the surrogate $\Phi$-risk $\mathcal{R}_\Phi$ via the so-called *calibration function*, see Definition 1 below [5, 49, 43, 18, 3]. As it is standard for this kind of analysis, the setup is *non-parametric*, i.e. it does not take into account the dependency of scores on input variables $\boldsymbol{x}$. For now, we assume that a family of score functions $\mathfrak{F}_\mathcal{F}$ consists of all vector-valued Borel measurable functions $\mathfrak{f} : \mathcal{X} \to \mathcal{F}$ where $\mathcal{F} \subseteq \mathbb{R}^k$ is a subspace of allowed score vectors, which will play an important role in our analysis. This setting is equivalent to a pointwise analysis, i.e, looking at the different input $\boldsymbol{x}$ independently. We bring the dependency on the input back into the analysis in Section 3.3 where we assume a specific family of score functions.

Let $\mathcal{D}_\mathcal{X}$ represent the marginal distribution for $\mathcal{D}$ on $\boldsymbol{x}$ and $\mathbf{P}(\cdot \mid \boldsymbol{x})$ denote its conditional given $\boldsymbol{x}$. We can now rewrite the risk $\mathcal{R}_L$ and $\Phi$-risk $\mathcal{R}_\Phi$ as

$$\mathcal{R}_L(\mathfrak{f}) = \mathbb{E}_{\boldsymbol{x} \sim \mathcal{D}_\mathcal{X}} \; \ell(\mathfrak{f}(\boldsymbol{x}), \mathbf{P}(\cdot \mid \boldsymbol{x})), \quad \mathcal{R}_\Phi(\mathfrak{f}) = \mathbb{E}_{\boldsymbol{x} \sim \mathcal{D}_\mathcal{X}} \; \phi(\mathfrak{f}(\boldsymbol{x}), \mathbf{P}(\cdot \mid \boldsymbol{x})),$$

where the conditional risk $\ell$ and the conditional $\Phi$-risk $\phi$ depend on a vector of scores $\boldsymbol{f}$ and a conditional distribution on the set of output labels $\boldsymbol{q}$ as

$$\ell(\boldsymbol{f}, \boldsymbol{q}) := \sum_{c=1}^{k} q_c L(\text{pred}(\boldsymbol{f}), c), \quad \phi(\boldsymbol{f}, \boldsymbol{q}) := \sum_{c=1}^{k} q_c \Phi(\boldsymbol{f}, c).$$

The *calibration function* $H_{\Phi,L,\mathcal{F}}$ between the surrogate loss $\Phi$ and the task loss $L$ relates the excess surrogate risk with the actual excess risk via the *excess risk bound*:

$$H_{\Phi,L,\mathcal{F}}(\delta\ell(\boldsymbol{f}, \boldsymbol{q})) \leq \delta\phi(\boldsymbol{f}, \boldsymbol{q}), \; \forall \boldsymbol{f} \in \mathcal{F}, \; \forall \boldsymbol{q} \in \Delta_k, \tag{4}$$

where $\delta\phi(\boldsymbol{f}, \boldsymbol{q}) = \phi(\boldsymbol{f}, \boldsymbol{q}) - \inf_{\hat{\boldsymbol{f}} \in \mathcal{F}} \phi(\hat{\boldsymbol{f}}, \boldsymbol{q})$, $\delta\ell(\boldsymbol{f}, \boldsymbol{q}) = \ell(\boldsymbol{f}, \boldsymbol{q}) - \inf_{\hat{\boldsymbol{f}} \in \mathcal{F}} \ell(\hat{\boldsymbol{f}}, \boldsymbol{q})$ are the excess risks and $\Delta_k$ denotes the probability simplex on $k$ elements.

In other words, to find a vector $\boldsymbol{f}$ that yields an excess risk smaller than $\varepsilon$, we need to optimize the $\Phi$-risk up to $H_{\Phi,L,\mathcal{F}}(\varepsilon)$ accuracy (in the worst case). We make this statement precise in Theorem 2 below, and now proceed to the formal definition of the calibration function.

**Definition 1** (Calibration function). *For a task loss $L$, a surrogate loss $\Phi$, a set of feasible scores $\mathcal{F}$, the* calibration function $H_{\Phi,L,\mathcal{F}}(\varepsilon)$ *(defined for $\varepsilon \geq 0$) equals the infimum excess of the conditional surrogate risk when the excess of the conditional actual risk is at least $\varepsilon$:*

$$H_{\Phi,L,\mathcal{F}}(\varepsilon) := \inf_{\boldsymbol{f} \in \mathcal{F}, \; \boldsymbol{q} \in \Delta_k} \delta\phi(\boldsymbol{f}, \boldsymbol{q}) \tag{5}$$

$$\text{s.t.} \quad \delta\ell(\boldsymbol{f}, \boldsymbol{q}) \geq \varepsilon. \tag{6}$$

*We set $H_{\Phi,L,\mathcal{F}}(\varepsilon)$ to $+\infty$ when the feasible set is empty.*

By construction, $H_{\Phi,L,\mathcal{F}}$ is non-decreasing on $[0, +\infty)$, $H_{\Phi,L,\mathcal{F}}(\varepsilon) \geq 0$, the inequality (4) holds, and $H_{\Phi,L,\mathcal{F}}(0) = 0$. Note that $H_{\Phi,L,\mathcal{F}}$ can be non-convex and even non-continuous (see examples in Figure 1). Also, note that large values of $H_{\Phi,L,\mathcal{F}}(\varepsilon)$ are better.

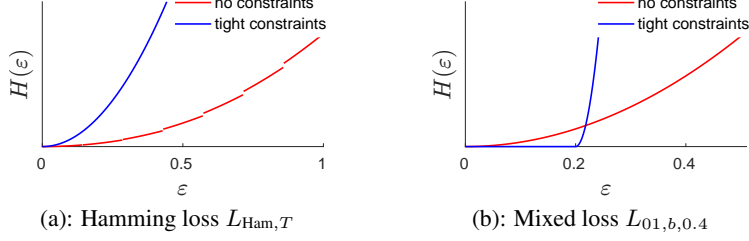

(a): Hamming loss $L_{\text{Ham},T}$        (b): Mixed loss $L_{01,b,0.4}$

Figure 1: Calibration functions for the quadratic surrogate $\Phi_{\text{quad}}$ (12) defined in Section 4 and two different task losses. (a) – the calibration functions for the Hamming loss $L_{\text{Ham},T}$ when used without constraints on the scores, $\mathcal{F} = \mathbb{R}^k$ (in red), and with the tight constraints implying consistency, $\mathcal{F} = \text{span}(L_{\text{Ham},T})$ (in blue). The red curve can grow exponentially slower than the blue one. (b) – the calibration functions for the mixed loss $L_{01,b,\eta}$ with $\eta = 0.4$ (see Section 2 for the definition) when used without constraints on the scores (red) and with tight constraints for the block 0-1 loss (blue). The blue curve represents level-0.2 consistency. The calibration function equals zero for $\varepsilon \leq \eta/2$, but grows exponentially faster than the red curve representing a consistent approach and thus could be better for small $\eta$. More details on the calibration functions in this figure are given in Section 4.

## 3.2    Notion of consistency

We use the calibration function $H_{\Phi,L,\mathcal{F}}$ to set a connection between optimizing the surrogate and task losses by Theorem 2, which is similar to Theorem 3 of Zhang [49].

**Theorem 2** (Calibration connection). *Let $H_{\Phi,L,\mathcal{F}}$ be the calibration function between the surrogate loss $\Phi$ and the task loss $L$ with feasible set of scores $\mathcal{F} \subseteq \mathbb{R}^k$. Let $\check{H}_{\Phi,L,\mathcal{F}}$ be a convex non-decreasing lower bound of the calibration function. Assume that $\Phi$ is continuous and bounded from below. Then, for any $\varepsilon > 0$ with finite $\check{H}_{\Phi,L,\mathcal{F}}(\varepsilon)$ and any $\mathfrak{f} \in \mathfrak{F}_{\mathcal{F}}$, we have*

$$\mathcal{R}_\Phi(\mathfrak{f}) < \mathcal{R}^*_{\Phi,\mathcal{F}} + \check{H}_{\Phi,L,\mathcal{F}}(\varepsilon) \;\; \Rightarrow \;\; \mathcal{R}_L(\mathfrak{f}) < \mathcal{R}^*_{L,\mathcal{F}} + \varepsilon, \tag{7}$$

*where $\mathcal{R}^*_{\Phi,\mathcal{F}} := \inf_{\mathfrak{f} \in \mathfrak{F}_{\mathcal{F}}} \mathcal{R}_\Phi(\mathfrak{f})$ and $\mathcal{R}^*_{L,\mathcal{F}} := \inf_{\mathfrak{f} \in \mathfrak{F}_{\mathcal{F}}} \mathcal{R}_L(\mathfrak{f})$.*

*Proof.* We take the expectation of (4) w.r.t. $\boldsymbol{x}$, where the second argument of $\ell$ is set to the conditional distribution $\mathbf{P}(\cdot \mid \boldsymbol{x})$. Then, we apply Jensen's inequality (since $\check{H}_{\Phi,L,\mathcal{F}}$ is convex) to get

$$\check{H}_{\Phi,L,\mathcal{F}}(\mathcal{R}_L(\mathfrak{f}) - \mathcal{R}^*_{L,\mathcal{F}}) \leq \mathcal{R}_\Phi(\mathfrak{f}) - \mathcal{R}^*_{\Phi,\mathcal{F}} < \check{H}_{\Phi,L,\mathcal{F}}(\varepsilon), \tag{8}$$

which implies (7) by monotonicity of $\check{H}_{\Phi,L,\mathcal{F}}$.      $\square$

A suitable convex non-decreasing lower bound $\check{H}_{\Phi,L,\mathcal{F}}(\varepsilon)$ required by Theorem 2 always exists, e.g., the zero constant. However, in this case Theorem 2 is not informative, because the l.h.s. of (7) is never true. Zhang [49, Proposition 25] claims that $\check{H}_{\Phi,L,\mathcal{F}}$ defined as the lower convex envelope of the calibration function $H_{\Phi,L,\mathcal{F}}$ satisfies $\check{H}_{\Phi,L,\mathcal{F}}(\varepsilon) > 0$, $\forall \varepsilon > 0$, if $H_{\Phi,L,\mathcal{F}}(\varepsilon) > 0$, $\forall \varepsilon > 0$, and, e.g., the set of labels is finite. This statement implies that an informative $\check{H}_{\Phi,L,\mathcal{F}}$ always exists and allows to characterize consistency through properties of the calibration function $H_{\Phi,L,\mathcal{F}}$.

We now define a notion of *level-$\eta$ consistency*, which is more general than consistency.

**Definition 3** (level-$\eta$ consistency). *A surrogate loss $\Phi$ is consistent up to level $\eta \geq 0$ w.r.t. a task loss $L$ and a set of scores $\mathcal{F}$ if and only if the calibration function satisfies $H_{\Phi,L,\mathcal{F}}(\varepsilon) > 0$ for all $\varepsilon > \eta$ and there exists $\hat{\varepsilon} > \eta$ such that $H_{\Phi,L,\mathcal{F}}(\hat{\varepsilon})$ is finite.*

Looking solely at (standard level-0) consistency vs. inconsistency might be too coarse to capture practical properties related to optimization accuracy (see, e.g., [29]). For example, if $H_{\Phi,L,\mathcal{F}}(\varepsilon) = 0$ only for very small values of $\varepsilon$, then the method can still optimize the actual risk up to a certain level which might be good enough in practice, especially if it means that it can be optimized faster. Examples of calibration functions for consistent and inconsistent surrogate losses are shown in Figure 1.

**Other notions of consistency.** Definition 3 with $\eta = 0$ and $\mathcal{F} = \mathbb{R}^k$ results in the standard setting often appearing in the literature. In particular, in this case Theorem 2 implies Fisher consistency as

formulated, e.g., by Pedregosa et al. [37] for general losses and Lin [27] for binary classification. This setting is also closely related to many definitions of consistency used in the literature. For example, for a bounded from below and continuous surrogate, it is equivalent to infinite-sample consistency [49], classification calibration [46], edge-consistency [18], $(L, \mathbb{R}^k)$-calibration [39], prediction calibration [48]. See [49, Appendix A] for the detailed discussion.

**Role of $\mathcal{F}$.** Let the *approximation error* for the restricted set of scores $\mathcal{F}$ be defined as $\mathcal{R}_{L,\mathcal{F}}^* - \mathcal{R}_L^* :=$ $\inf_{\mathfrak{f} \in \mathfrak{F}_{\mathcal{F}}} \mathcal{R}_L(\mathfrak{f}) - \inf_{\mathfrak{f}} \mathcal{R}_L(\mathfrak{f})$. For any conditional distribution $q$, the score vector $\boldsymbol{f} := -Lq$ will yield an optimal prediction. Thus the condition $\operatorname{span}(L) \subseteq \mathcal{F}$ is sufficient for $\mathcal{F}$ to have zero approximation error for any distribution $\mathcal{D}$, and for our 0-consistency condition to imply the standard Fisher consistency with respect to $L$. In the following, we will see that a restricted $\mathcal{F}$ can both play a role for computational efficiency as well as statistical efficiency (thus losses with smaller $\operatorname{span}(L)$ might be easier to work with).

### 3.3 Connection to optimization accuracy and statistical efficiency

The scale of a calibration function is not intrinsically well-defined: we could multiply the surrogate function by a scalar and it would multiply the calibration function by the same scalar, without changing the optimization problem. Intuitively, we would like the surrogate loss to be of order $1$. If with this scale the calibration function is exponentially small (has a $1/k$ factor), then we have strong evidence that the stochastic optimization will be difficult (and thus learning will be slow).

To formalize this intuition, we add to the picture the *complexity* of optimizing the surrogate loss with a *stochastic approximation* algorithm. By using a scale-invariant convergence rate, we provide a natural normalization of the calibration function. The following two observations are central to the theoretical insights provided in our work:

1. **Scale.** For a properly scaled surrogate loss, the *scale* of the calibration function is a good indication of whether a stochastic approximation algorithm will take a large number of iterations (in the worst case) to obtain guarantees of small excess of the actual risk (and vice-versa, a large coefficient indicates a small number of iterations). The actual verification requires computing the normalization quantities given in Theorem 6 below.

2. **Statistics.** The bound on the number of iterations directly relates to the number of training examples that would be needed to learn, if we see each iteration of the stochastic approximation algorithm as using one training example to optimize the expected surrogate.

To analyze the statistical convergence of surrogate risk optimization, we have to specify the set of score functions that we work with. We assume that the structure on input $\boldsymbol{x} \in \mathcal{X}$ is defined by a positive definite kernel $K : \mathcal{X} \times \mathcal{X} \to \mathbb{R}$. We denote the corresponding reproducing kernel Hilbert space (RKHS) by $\mathcal{H}$ and its explicit feature map by $\psi(\boldsymbol{x}) \in \mathcal{H}$. By the reproducing property, we have $\langle f, \psi(\boldsymbol{x}) \rangle_{\mathcal{H}} = f(\boldsymbol{x})$ for all $\boldsymbol{x} \in \mathcal{X}$, $f \in \mathcal{H}$, where $\langle \cdot, \cdot \rangle_{\mathcal{H}}$ is the inner product in the RKHS. We define the subspace of allowed scores $\mathcal{F} \subseteq \mathbb{R}^k$ via the span of the columns of a matrix $F \in \mathbb{R}^{k \times r}$. The matrix $F$ explicitly defines the structure of the score function. With this notation, we will assume that the score function is of the form $\mathfrak{f}(\boldsymbol{x}) = FW\psi(\boldsymbol{x})$, where $W : \mathcal{H} \to \mathbb{R}^r$ is a linear operator to be learned (a matrix if $\mathcal{H}$ is of finite dimension) that represents a collection of $r$ elements in $\mathcal{H}$, transforming $\psi(\boldsymbol{x})$ to a vector in $\mathbb{R}^r$ by applying the RKHS inner product $r$ times.[3] Note that for structured losses, we usually have $r \ll k$. The set of all score functions is thus obtained by varying $W$ in this definition and is denoted by $\mathfrak{F}_{F,\mathcal{H}}$. As a concrete example of a score family $\mathfrak{F}_{F,\mathcal{H}}$ for structured prediction, consider the standard sequence model with unary and pairwise potentials. In this case, the dimension $r$ equals $Ts + (T-1)s^2$, where $T$ is the sequence length and $s$ is the number of labels of each variable. The columns of the matrix $F$ consist of $2T - 1$ groups (one for each unary and pairwise potential). Each row of $F$ has exactly one entry equal to one in each column group (with zeros elsewhere).

In this setting, we use the online projected averaged stochastic subgradient descent ASGD[4] (stochastic w.r.t. data $(\boldsymbol{x}^{(n)}, \boldsymbol{y}^{(n)}) \sim \mathcal{D}$) to minimize the surrogate risk directly [6]. The $n$-th update consists in

$$W^{(n)} := P_D\big[W^{(n-1)} - \gamma^{(n)} F^{\mathsf{T}} \nabla \Phi \psi(\boldsymbol{x}^{(n)})^{\mathsf{T}}\big], \tag{9}$$

where $F^\mathsf{T}\nabla\Phi\psi(\boldsymbol{x}^{(n)})^\mathsf{T} : \mathcal{H} \to \mathbb{R}^r$ is the stochastic functional gradient, $\gamma^{(n)}$ is the step size and $P_D$ is the projection on the ball of radius $D$ w.r.t. the Hilbert–Schmidt norm.[5] The vector $\nabla\Phi \in \mathbb{R}^k$ is a regular gradient of the sampled surrogate $\Phi(\mathfrak{f}(\boldsymbol{x}^{(n)}), \boldsymbol{y}^{(n)})$ w.r.t. the scores, $\nabla\Phi = \nabla_{\boldsymbol{f}}\Phi(\boldsymbol{f}, \boldsymbol{y}^{(n)})|_{\boldsymbol{f}=\mathfrak{f}(\boldsymbol{x}^{(n)})}$. We wrote the above update using an explicit feature map $\psi$ for notational simplicity, but kernel ASGD can also be implemented without it by using the kernel trick. The convergence properties of ASGD in RKHS are analogous to the finite-dimensional ASGD because they rely on dimension-free quantities. To use a simple convergence analysis, we follow Ciliberto et al. [10] and make the following simplifying assumption:

**Assumption 4** (Well-specified optimization w.r.t. the function class $\mathfrak{F}_{F,\mathcal{H}}$). *The distribution $\mathcal{D}$ is such that $\mathcal{R}^*_{\Phi,\mathcal{F}} := \inf_{\mathfrak{f}\in\mathfrak{F}_\mathcal{F}} \mathcal{R}_\Phi(\mathfrak{f})$ has some global minimum $\mathfrak{f}^*$ that also belongs to $\mathfrak{F}_{F,\mathcal{H}}$.*

Assumption 4 simply means that each row of $W^*$ defining $\mathfrak{f}^*$ belongs to the RKHS $\mathcal{H}$ implying a finite norm $\|W^*\|_{HS}$. Assumption 4 can be relaxed if the kernel $K$ is universal, but then the convergence analysis becomes much more complicated [36].

**Theorem 5** (Convergence rate). *Under Assumption 4 and assuming that (i) the functions $\Phi(\boldsymbol{f}, \boldsymbol{y})$ are bounded from below and convex w.r.t. $\boldsymbol{f} \in \mathbb{R}^k$ for all $\boldsymbol{y} \in \mathcal{Y}$; (ii) the expected square of the norm of the stochastic gradient is bounded, $\mathbf{E}_{(\boldsymbol{x},\boldsymbol{y})\sim\mathcal{D}}\|F^\mathsf{T}\nabla\Phi\psi(\boldsymbol{x})^\mathsf{T}\|^2_{HS} \leq M^2$ and (iii) $\|W^*\|_{HS} \leq D$, then running the ASGD algorithm (9) with the constant step-size $\gamma := \frac{2D}{M\sqrt{N}}$ for $N$ steps admits the following expected suboptimality for the averaged iterate $\bar{\mathfrak{f}}^{(N)}$:*

$$\mathbf{E}[\mathcal{R}_\Phi(\bar{\mathfrak{f}}^{(N)})] - \mathcal{R}^*_{\Phi,\mathcal{F}} \leq \frac{2DM}{\sqrt{N}} \quad \text{where} \quad \bar{\mathfrak{f}}^{(N)} := \frac{1}{N}\sum_{n=1}^N FW^{(n)}\psi(\boldsymbol{x}^{(n)})^\mathsf{T}. \tag{10}$$

Theorem 5 is a straight-forward extension of classical results [33, 36].

By combining the convergence rate of Theorem 5 with Theorem 2 that connects the surrogate and actual risks, we get Theorem 6 which explicitly gives the number of iterations required to achieve $\varepsilon$ accuracy on the expected *population risk* (see App. A for the proof). Note that since ASGD is applied in an online fashion, Theorem 6 also serves as the sample complexity bound, i.e., says how many samples are needed to achieve $\varepsilon$ target accuracy (compared to the best prediction rule if $\mathcal{F}$ has zero approximation error).

**Theorem 6** (Learning complexity). *Under the assumptions of Theorem 5, for any $\varepsilon > 0$, the random (w.r.t. the observed training set) output $\bar{\mathfrak{f}}^{(N)} \in \mathfrak{F}_{F,\mathcal{H}}$ of the ASGD algorithm after*

$$N > N^* := \frac{4D^2M^2}{\check{H}^2_{\Phi,L,\mathcal{F}}(\varepsilon)} \tag{11}$$

*iterations has the expected excess risk bounded with $\varepsilon$, i.e., $\mathbf{E}[\mathcal{R}_L(\bar{\mathfrak{f}}^{(N)})] < \mathcal{R}^*_{L,\mathcal{F}} + \varepsilon$.*

## 4 Calibration function analysis for quadratic surrogate

A major challenge to applying Theorem 6 is the computation of the calibration function $H_{\Phi,L,\mathcal{F}}$. In App. C, we present a generalization to arbitrary multi-class losses of a surrogate loss class from Zhang [49, Section 4.4.2] that is consistent for any task loss $L$. Here, we consider the simplest example of this family, called the *quadratic surrogate* $\Phi_{\text{quad}}$, which has the advantage that we can bound or even compute exactly its calibration function. We define the quadratic surrogate as

$$\Phi_{\text{quad}}(\boldsymbol{f}, \boldsymbol{y}) := \frac{1}{2k}\|\boldsymbol{f} + L(:,\boldsymbol{y})\|^2_2 = \frac{1}{2k}\sum_{c=1}^k (f_c^2 + 2f_c L(c, \boldsymbol{y}) + L(c, \boldsymbol{y})^2). \tag{12}$$

One simple sufficient condition for the surrogate (12) to be consistent and also to have zero approximation error is that $\mathcal{F}$ fully contains $\text{span}(L)$. To make the dependence on the score subspace explicit, we parameterize it with a matrix $F \in \mathbb{R}^{k\times r}$ with the number of columns $r$ typically being much smaller than the number of labels $k$. With this notation, we have $\mathcal{F} = \text{span}(F) = \{F\boldsymbol{\theta} \mid \boldsymbol{\theta} \in \mathbb{R}^r\}$, and the dimensionality of $\mathcal{F}$ equals the rank of $F$, which is at most $r$.[6]

For the quadratic surrogate (12), the excess of the expected surrogate takes a simple form:

$$\delta\phi_{\text{quad}}(F\boldsymbol{\theta}, \boldsymbol{q}) = \tfrac{1}{2k}\|F\boldsymbol{\theta} + L\boldsymbol{q}\|_2^2. \tag{13}$$

Equation (13) holds under the assumption that the subspace $\mathcal{F}$ contains the column space of the loss matrix $\text{span}(L)$, which also means that the set $\mathcal{F}$ contains the optimal prediction for any $\boldsymbol{q}$ (see Lemma 9 in App. B for the proof). Importantly, the function $\delta\phi_{\text{quad}}(F\boldsymbol{\theta}, \boldsymbol{q})$ is jointly convex in the conditional probability $\boldsymbol{q}$ and parameters $\boldsymbol{\theta}$, which simplifies its analysis.

**Lower bound on the calibration function.** We now present our main technical result: a lower bound on the calibration function for the surrogate loss $\Phi_{\text{quad}}$ (12). This lower bound characterizes the easiness of learning with this surrogate given the scaling intuition mentioned in Section 3.3. The proof of Theorem 7 is given in App. D.1.

**Theorem 7** (Lower bound on $H_{\Phi_{\text{quad}}}$). *For any task loss $L$, its quadratic surrogate $\Phi_{\text{quad}}$, and a score subspace $\mathcal{F}$ containing the column space of $L$, the calibration function can be lower bounded:*

$$H_{\Phi_{\text{quad}}, L, \mathcal{F}}(\varepsilon) \geq \frac{\varepsilon^2}{2k \max_{i \neq j}\|P_{\mathcal{F}}\Delta_{ij}\|_2^2} \geq \frac{\varepsilon^2}{4k}, \tag{14}$$

*where $P_{\mathcal{F}}$ is the orthogonal projection on the subspace $\mathcal{F}$ and $\Delta_{ij} = \mathbf{e}_i - \mathbf{e}_j \in \mathbb{R}^k$ with $\mathbf{e}_c$ being the c-th basis vector of the standard basis in $\mathbb{R}^k$.*

**Lower bound for specific losses.** We now discuss the meaning of the bound (14) for some specific losses (the detailed derivations are given in App. D.3). For the 0-1, block 0-1 and Hamming losses ($L_{01}$, $L_{01,b}$ and $L_{\text{Ham},T}$, respectively) with the smallest possible score subspaces $\mathcal{F}$, the bound (14) gives $\frac{\varepsilon^2}{4k}$, $\frac{\varepsilon^2}{4b}$ and $\frac{\varepsilon^2}{8T}$, respectively. All these bounds are tight (see App. E). However, if $\mathcal{F} = \mathbb{R}^k$ the bound (14) is not tight for the block 0-1 and mixed losses (see also App. E). In particular, the bound (14) cannot detect level-$\eta$ consistency for $\eta > 0$ (see Def. 3) and does not change when the loss changes, but the score subspace stays the same.

**Upper bound on the calibration function.** Theorem 8 below gives an upper bound on the calibration function holding for unconstrained scores, i.e, $\mathcal{F} = \mathbb{R}^k$ (see the proof in App. D.2). This result shows that without some appropriate constraints on the scores, efficient learning is not guaranteed (in the worst case) because of the $1/k$ scaling of the calibration function.

**Theorem 8** (Upper bound on $H_{\Phi_{\text{quad}}}$). *If a loss matrix $L$ with $L_{max} > 0$ defines a pseudometric[7] on labels and there are no constraints on the scores, i.e., $\mathcal{F} = \mathbb{R}^k$, then the calibration function for the quadratic surrogate $\Phi_{\text{quad}}$ can be upper bounded: $H_{\Phi_{\text{quad}}, L, \mathbb{R}^k}(\varepsilon) \leq \frac{\varepsilon^2}{2k}, \quad 0 \leq \varepsilon \leq L_{max}$.*

From our lower bound in Theorem 7 (which guarantees consistency), the natural constraint on the score is $\mathcal{F} = \text{span}(L)$, with the dimension of this space giving an indication of the intrinsic "difficulty" of a loss. Computations for the lower bounds in some specific cases (see App. D.3 for details) show that the 0-1 loss is "hard" while the block 0-1 loss and the Hamming loss are "easy". Note that in all these cases the lower bound (14) is tight, see the discussion below.

**Exact calibration functions.** Note that the bounds proven in Theorems 7 and 8 imply that, in the case of no constraints on the scores $\mathcal{F} = \mathbb{R}^k$, for the 0-1, block 0-1 and Hamming losses, we have

$$\frac{\varepsilon^2}{4k} \leq H_{\Phi_{\text{quad}}, L, \mathbb{R}^k}(\varepsilon) \leq \frac{\varepsilon^2}{2k}, \tag{15}$$

where $L$ is the matrix defining a loss. For completeness, in App. E, we compute the exact calibration functions for the 0-1 and block 0-1 losses. Note that the calibration function for the **0-1 loss** equals the lower bound, illustrating the worst-case scenario. To get some intuition, an example of a conditional distribution $\boldsymbol{q}$ that gives the (worst case) value to the calibration function (for several losses) is $q_i = \frac{1}{2} + \frac{\varepsilon}{2}$, $q_j = \frac{1}{2} - \frac{\varepsilon}{2}$ and $q_c = 0$ for $c \notin \{i, j\}$. See the proof of Proposition 12 in App. E.1.

In what follows, we provide the calibration functions in the cases with constraints on the scores. For the **block 0-1 loss** with $b$ equal blocks and under constraints that the scores within blocks are equal, the calibration function equals (see Proposition 14 of App. E.2)

$$H_{\Phi_{\text{quad}}, L_{01,b}, \mathcal{F}_{01,b}}(\varepsilon) = \frac{\varepsilon^2}{4b}, \quad 0 \leq \varepsilon \leq 1. \tag{16}$$

For the **Hamming loss** defined over $T$ binary variables and under constraints implying separable scores, the calibration function equals (see Proposition 15 in App. E.3)

$$H_{\Phi_{\text{quad}}, L_{\text{Ham},T}, \mathcal{F}_{\text{Ham},T}}(\varepsilon) = \frac{\varepsilon^2}{8T}, \ 0 \leq \varepsilon \leq 1. \tag{17}$$

The calibration functions (16) and (17) depend on the quantities representing the actual complexities of the loss (the number of blocks $b$ and the length of the sequence $T$) and can be exponentially larger than the upper bound for the unconstrained case.

In the case of **mixed 0-1 and block 0-1 loss**, if the scores $\boldsymbol{f}$ are constrained to be equal inside the blocks, i.e., belong to the subspace $\mathcal{F}_{01,b} = \text{span}(L_{01,b}) \subsetneq \mathbb{R}^k$, then the calibration function is equal to 0 for $\varepsilon \leq \frac{\eta}{2}$, implying inconsistency (and also note that the approximation error can be as big as $\eta$ for $\mathcal{F}_{01,b}$). However, for $\varepsilon > \frac{\eta}{2}$, the calibration function is of the order $\frac{1}{b}(\varepsilon - \frac{\eta}{2})^2$. See Figure 1b for the illustration of this calibration function and Proposition 17 of App. E.4 for the exact formulation and the proof. Note that while the calibration function for the constrained case is inconsistent, its value can be exponentially larger than the one for the unconstrained case for $\varepsilon$ big enough and when the blocks are exponentially large (see Proposition 16 of App. E.4).

**Computation of the SGD constants.** Applying the learning complexity Theorem 6 requires to compute the quantity $DM$ where $D$ bounds the norm of the optimal solution and $M$ bounds the expected square of the norm of the stochastic gradient. In App. F, we provide a way to bound this quantity for our quadratic surrogate (12) under the simplifying assumption that each conditional $q_c(\boldsymbol{x})$ (seen as function of $\boldsymbol{x}$) belongs to the RKHS $\mathcal{H}$ (which implies Assumption 4). In particular, we get

$$DM = L_{\max}^2 \xi(\kappa(F)\sqrt{r}RQ_{\max}), \quad \xi(z) = z^2 + z, \tag{18}$$

where $\kappa(F)$ is the condition number of the matrix $F$, $R$ is an upper bound on the RKHS norm of object feature maps $\|\psi(\boldsymbol{x})\|_{\mathcal{H}}$. We define $Q_{\max}$ as an upper bound on $\sum_{c=1}^{k} \|q_c\|_{\mathcal{H}}$ (can be seen as the generalization of the inequality $\sum_{c=1}^{k} q_c \leq 1$ for probabilities). The constants $R$ and $Q_{\max}$ depend on the data, the constant $L_{\max}$ depends on the loss, $r$ and $\kappa(F)$ depend on the choice of matrix $F$.

We compute the constant $DM$ for the specific losses that we considered in App. F.1. For the 0-1, block 0-1 and Hamming losses, we have $DM = O(k)$, $DM = O(b)$ and $DM = O(\log_2^3 k)$, respectively. These computations indicate that the quadratic surrogate allows efficient learning for structured block 0-1 and Hamming losses, but that the convergence could be slow in the worst case for the 0-1 loss.

## 5 Related works

**Consistency for multi-class problems.** Building on significant progress for the case of binary classification, see, e.g. [5], there has been a lot of interest in the multi-class case. Zhang [49] and Tewari & Bartlett [46] analyze the consistency of many existing surrogates for the 0-1 loss. Gao & Zhou [20] focus on multi-label classification. Narasimhan et al. [32] provide a consistent algorithm for arbitrary multi-class loss defined by a function of the confusion matrix. Recently, Ramaswamy & Agarwal [39] introduce the notion of convex calibrated dimension, as the minimal dimensionality of the score vector that is required for consistency. In particular, they showed that for the Hamming loss on $T$ binary variables, this dimension is at most $T$. In our analysis, we use scores of rank $(T+1)$, see (35) in App. D.3, yielding a similar result.

The task of ranking has attracted a lot of attention and [18, 8, 9, 40] analyze different families of surrogate and task losses proving their (in-)consistency. In this line of work, Ramaswamy et al. [40] propose a quadratic surrogate for an arbitrary low rank loss which is related to our quadratic surrogate (12). They also prove that several important ranking losses, i.e., precision@q, expected rank utility, mean average precision and pairwise disagreement, are of low-rank. We conjecture that our approach is compatible with these losses and leave precise connections as future work.

**Structured SVM (SSVM) and friends.** SSVM [44, 45, 47] is one of the most used convex surrogates for tasks with structured outputs, thus, its consistency has been a question of great interest. It is known that Crammer-Singer multi-class SVM [15], which SSVM is built on, is not consistent for 0-1 loss unless there is a majority class with probability at least $\frac{1}{2}$ [49, 31]. However, it is consistent for the "abstain" and ordinal losses in the case of 3 classes [39]. Structured ramp loss and probit surrogates are closely related to SSVM and are consistent [31, 16, 30, 23], but not convex.

Recently, Doğan et al. [17] categorized different versions of multi-class SVM and analyzed them from Fisher and universal consistency point of views. In particular, they highlight differences between Fisher and universal consistency and give examples of surrogates that are Fisher consistent, but not universally consistent and vice versa. They also highlight that the Crammer-Singer SVM is neither Fisher, not universally consistent even with a careful choice of regularizer.

**Quadratic surrogates for structured prediction.** Ciliberto et al. [10] and Brouard et al. [7] consider minimizing $\sum_{i=1}^{n} \|g(\boldsymbol{x}_i) - \psi_o(\boldsymbol{y}_i)\|_{\mathcal{H}}^2$ aiming to match the RKHS embedding of inputs $g : \mathcal{X} \to \mathcal{H}$ to the feature maps of outputs $\psi_o : \mathcal{Y} \to \mathcal{H}$. In their frameworks, the task loss is not considered at the learning stage, but only at the prediction stage. Our quadratic surrogate (12) depends on the loss directly. The empirical risk defined by both their and our objectives can be minimized analytically with the help of the kernel trick and, moreover, the resulting predictors are identical. However, performing such computation in the case of large dataset can be intractable and the generalization properties have to be taken care of, e.g., by the means of regularization. In the large-scale scenario, it is more natural to apply stochastic optimization (e.g., kernel ASGD) that directly minimizes the population risk and has better dependency on the dataset size. When combined with stochastic optimization, the two approaches lead to different behavior. In our framework, we need to estimate $r = \text{rank}(L)$ scalar functions, but the alternative needs to estimate $k$ functions (if, e.g., $\psi_o(\boldsymbol{y}) = \mathbf{e}_{\boldsymbol{y}} \in \mathbb{R}^k$), which results in significant differences for low-rank losses, such as block 0-1 and Hamming.

**Calibration functions.** Bartlett et al. [5] and Steinwart [43] provide calibration functions for most existing surrogates for binary classification. All these functions differ in term of shape, but are roughly similar in terms of constants. Pedregosa et al. [37] generalize these results to the case of ordinal regression. However, their calibration functions have at best a $1/k$ factor if the surrogate is normalized w.r.t. the number of classes. The task of ranking has been of significant interest. However, most of the literature [e.g., 11, 14, 24, 1], only focuses on calibration functions (in the form of regret bounds) for bipartite ranking, which is more akin to cost-sensitive binary classification.

Ávila Pires et al. [3] generalize the theoretical framework developed by Steinwart [43] and present results for the multi-class SVM of Lee et al. [26] (the score vectors are constrained to sum to zero) that can be built for any task loss of interest. Their surrogate $\Phi$ is of the form $\sum_{c \in \mathcal{Y}} L(c, \boldsymbol{y}) a(f_c)$ where $\sum_{c \in \mathcal{Y}} f_c = 0$ and $a(f)$ is some convex function with all subgradients at zero being positive. The recent work by Ávila Pires & Szepesvári [2] refines the results, but specifically for the case of 0-1 loss. In this line of work, the surrogate is typically not normalized by $k$, and if normalized the calibration functions have the constant $1/k$ appearing.

Finally, Ciliberto et al. [10] provide the calibration function for their quadratic surrogate. Assuming that the loss can be represented as $L(\hat{\boldsymbol{y}}, \boldsymbol{y}) = \langle V \psi_o(\hat{\boldsymbol{y}}), \psi_o(\boldsymbol{y}) \rangle_{\mathcal{H}_{\mathcal{Y}}}, \hat{\boldsymbol{y}}, \boldsymbol{y} \in \mathcal{Y}$ (this assumption can always be satisfied in the case of a finite number of labels, by taking $V$ as the loss matrix $L$ and $\psi_o(\boldsymbol{y}) := \mathbf{e}_{\boldsymbol{y}} \in \mathbb{R}^k$ where $\mathbf{e}_{\boldsymbol{y}}$ is the $\boldsymbol{y}$-th vector of the standard basis in $\mathbb{R}^k$). In their Theorem 2, they provide an excess risk bound leading to a lower bound on the corresponding calibration function $H_{\Phi,L,\mathbb{R}^k}(\varepsilon) \geq \frac{\varepsilon^2}{c_\Delta^2}$ where a constant $c_\Delta = \|V\|_2 \max_{\boldsymbol{y} \in \mathcal{Y}} \|\psi_o(\boldsymbol{y})\|$ simply equals the spectral norm of the loss matrix for the finite-dimensional construction provided above. However, the spectral norm of the loss matrix is exponentially large even for highly structured losses such as the block 0-1 and Hamming losses, i.e., $\|L_{01,b}\|_2 = k - \frac{k}{b}, \|L_{\text{Ham},T}\|_2 = \frac{k}{2}$. This conclusion puts the objective of Ciliberto et al. [10] in line with ours when no constraints are put on the scores.

# 6   Conclusion

In this paper, we studied the consistency of convex surrogate losses specifically in the context of structured prediction. We analyzed calibration functions and proposed an optimization-based normalization aiming to connect consistency with the existence of efficient learning algorithms. Finally, we instantiated all components of our framework for several losses by computing the calibration functions and the constants coming from the normalization. By carefully monitoring exponential constants, we highlighted the difference between tractable and intractable task losses.

These were first steps in advancing our theoretical understanding of consistent structured prediction. Further steps include analyzing more losses such as the low-rank ranking losses studied by Ramaswamy et al. [40] and, instead of considering constraints on the scores, one could instead put constraints on the set of distributions to investigate the effect on the calibration function.

**Acknowledgements**

We would like to thank Pascal Germain for useful discussions. This work was partly supported by the ERC grant Activia (no. 307574), the NSERC Discovery Grant RGPIN-2017-06936 and the MSR-INRIA Joint Center.

## Footnotes

*DI École normale supérieure, CNRS, PSL Research University

†National Research University Higher School of Economics

[1]Our analysis is generalizable to rectangular losses, e.g., ranking losses studied by Ramaswamy et al. [40].

[2] Here we use the Iverson bracket notation, i.e., $[A] := 1$ if a logical expression $A$ is true, and zero otherwise.

[3]Note that if $\operatorname{rank}(F) = r$, our setup is equivalent to assuming a *joint kernel* [47] in the product form: $K_{\text{joint}}((\boldsymbol{x}, c), (\boldsymbol{x}', c')) := K(\boldsymbol{x}, \boldsymbol{x}')F(c, :)F(c', :)^{\mathsf{T}}$, where $F(c, :)$ is the row $c$ for matrix $F$.

[4]See, e.g., [36] for the formal setup of kernel ASGD.

[5]The Hilbert–Schmidt norm of a linear operator $A$ is defined as $\|A\|_{HS} = \sqrt{\text{tr}A^\ddagger A}$ where $A^\ddagger$ is the adjoint operator. In the case of finite dimension, the Hilbert–Schmidt norm coincides with the Frobenius matrix norm.

[6]Evaluating $\Phi_{\text{quad}}$ requires computing $F^\mathsf{T}F$ and $F^\mathsf{T}L(:,\boldsymbol{y})$ for which direct computation is intractable when $k$ is exponential, but which can be done in closed form for the structured losses we consider (the Hamming and block 0-1 loss). More generally, these operations require suitable inference algorithms. See also App. F.

[7]A pseudometric is a function $d(a, b)$ satisfying the following axioms: $d(x, y) \geq 0$, $d(x, x) = 0$ (but possibly $d(x, y) = 0$ for some $x \neq y$), $d(x, y) = d(y, x)$, $d(x, z) \leq d(x, y) + d(y, z)$.

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
