[Supplementary Material]

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

[8]Zhang [49] refers to this surrogate as "decoupled unconstrained background discriminative surrogate". Note the $^1/_k$ scaling to make $\Phi_{a,b}$ of order 1.

[9]To simplify the statement of Theorem 7, we removed the constraints $i, j \in \text{pred}(\mathcal{F})$ from Lemma 10 which said that we should consider only the labels that can be predicted with some feasible scores. A potentially tighter lower bound can be obtained by keeping the $i, j \in \text{pred}(\mathcal{F})$ constraint.

[10]The possibility $P_F \Delta_{ij} = 0$ is also covered by this equation with the convention that $1/0 = \infty$ (in this case, $\mu^* = \infty$).

[11] https://github.com/aosokin/consistentSurrogates_derivations

[12]Note that just showing the feasibility of the assigned values $\boldsymbol{q}^*$ and $\boldsymbol{f}^*$ give us an upper bound on the calibration function. In the case of the 0-1 loss, it appears that this upper bound matches the lower bound provided by Theorem 7, so we do not need to prove optimality explicitly. However, we still give this proof as a simple illustration of the proof technique as its structure will be re-used also for the cases when the bound of Theorem 7 is not tight.

[13]If these optimal scores are not equal, by symmetry, one can obtain the same objective and feasibility by permuting their corresponding values. By taking a uniform convex combination on all permutations, we obtain a point where all the scores are equal, and by convexity, would yield a lower objective value.

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

# Supplementary Material (Appendix)

# On Structured Prediction Theory with Calibrated Convex Surrogate Losses

## Outline

## A    Learning complexity theorem

**Theorem 6** (Learning complexity). *Under the assumptions of Theorem 5, for any $\varepsilon > 0$, the random (w.r.t. the observed training set) output $\bar{\mathfrak{f}}^{(N)} \in \mathfrak{F}_{F,\mathcal{H}}$ of the ASGD algorithm after*

$$N > N^* := \frac{4D^2 M^2}{\check{H}^2_{\Phi,L,\mathcal{F}}(\varepsilon)} \tag{19}$$

*iterations has the expected excess risk bounded with $\varepsilon$, i.e., $\mathbf{E}[\mathcal{R}_L(\bar{\mathfrak{f}}^{(N)})] < \mathcal{R}^*_{L,\mathcal{F}} + \varepsilon$.*

*Proof.* By (10) from Theorem 5, $N$ steps of the algorithm, in expectation, result in $\check{H}_{\Phi,L,\mathcal{F}}(\varepsilon)$ accuracy on the surrogate risk, i.e., $\mathbf{E}[\mathcal{R}_\Phi(\bar{\mathfrak{f}}^{(N)})] - \mathcal{R}^*_{\Phi,\mathcal{F}} < \check{H}_{\Phi,L,\mathcal{F}}(\varepsilon)$. We now generalize the proof of Theorem 2 to the case of expectation w.r.t. $\bar{\mathfrak{f}}^{(N)}$ depending on the random samples used by the ASGD algorithm. We take the expectation of (4) w.r.t. $\bar{\mathfrak{f}}^{(N)}$ substituted as $\mathfrak{f}$ and use Jensen's inequality (by convexity of $\check{H}_{\Phi,L,\mathcal{F}}$) to get $\mathbf{E}[\mathcal{R}_\Phi(\bar{\mathfrak{f}}^{(N)})] - \mathcal{R}^*_{\Phi,\mathcal{F}} \geq \mathbf{E}[\check{H}_{\Phi,L,\mathcal{F}}(\mathcal{R}_L(\bar{\mathfrak{f}}^{(N)}) - \mathcal{R}^*_{L,\mathcal{F}})] \geq \check{H}_{\Phi,L,\mathcal{F}}(\mathbf{E}[\mathcal{R}_L(\bar{\mathfrak{f}}^{(N)})] - \mathcal{R}^*_{L,\mathcal{F}})$. Finally, monotonicity of $\check{H}_{\Phi,L,\mathcal{F}}$ implies $\mathbf{E}[\mathcal{R}_L(\bar{\mathfrak{f}}^{(N)})] - \mathcal{R}^*_{L,\mathcal{F}} < \varepsilon$. $\qquad\square$

## B    Technical lemmas

In this section, we prove two technical lemmas that simplify the proofs of the main theoretical claims of the paper.

Lemma 9 computes the excess of the weighted surrogate risk $\delta\phi$ for the quadratic loss $\Phi_{\text{quad}}$ (12), which is central to our analysis presented in Section 4. The key property of this result is that the excess $\delta\phi$ is jointly convex w.r.t. the parameters $\boldsymbol{\theta}$ and conditional distribution $\boldsymbol{q}$, which simplifies further analysis.

Lemma 10 allows to cope with the combinatorial aspect of the computation of the calibration function. In particular, when the excess of the weighted surrogate risk is convex, Lemma 10 reduces the computation of the calibration function to a set of convex optimization problems, which often can be solved analytically. For symmetric losses, such as the 0-1, block 0-1 and Hamming losses, Lemma 10 also provides "symmetry breaking", meaning that many of the obtained convex optimization problems are identical up to a permutation of labels.

**Lemma 9.** *Consider the quadratic surrogate* $\Phi_{\text{quad}}$ *(12) defined for a task loss* $L$. *Let a subspace of scores* $\mathcal{F} \subseteq \mathbb{R}^k$ *be parametrized by* $\boldsymbol{\theta} \in \mathbb{R}^r$, *i.e.,* $\boldsymbol{f} = F\boldsymbol{\theta} \in \mathcal{F}$ *with* $F \in \mathbb{R}^{k \times r}$, *and assume that* $\text{span}(L) \subseteq \mathcal{F}$. *Then, the excess of the weighted surrogate loss can be expressed as*

$$\delta\phi_{\text{quad}}(F\boldsymbol{\theta}, \boldsymbol{q}) := \phi_{\text{quad}}(F\boldsymbol{\theta}, \boldsymbol{q}) - \inf_{\boldsymbol{\theta}' \in \mathbb{R}^r} \phi_{\text{quad}}(F\boldsymbol{\theta}', \boldsymbol{q}) = \tfrac{1}{2k}\|F\boldsymbol{\theta} + L\boldsymbol{q}\|_2^2.$$

*Proof.* By using the definition of the quadratic surrogate $\Phi_{\text{quad}}$ (12), we have

$$\phi(\boldsymbol{f}(\boldsymbol{\theta}), \boldsymbol{q}) = \tfrac{1}{2k}(\boldsymbol{\theta}^\mathsf{T} F^\mathsf{T} F \boldsymbol{\theta} + 2\boldsymbol{\theta}^\mathsf{T} F^\mathsf{T} L\boldsymbol{q}) + r(\boldsymbol{q}),$$
$$\boldsymbol{\theta}^* := \text{argmin}_{\boldsymbol{\theta}} \, \phi(\boldsymbol{f}(\boldsymbol{\theta}), \boldsymbol{q}) = -(F^\mathsf{T} F)^\dagger F^\mathsf{T} L\boldsymbol{q},$$
$$\delta\phi(\boldsymbol{f}(\boldsymbol{\theta}), \boldsymbol{q}) = \tfrac{1}{2k}(\boldsymbol{\theta}^\mathsf{T} F^\mathsf{T} F \boldsymbol{\theta} + 2\boldsymbol{\theta}^\mathsf{T} F^\mathsf{T} L\boldsymbol{q}$$
$$+ \boldsymbol{q}^\mathsf{T} L^\mathsf{T} F (F^\mathsf{T} F)^\dagger F^\mathsf{T} L\boldsymbol{q}),$$

where $r(\boldsymbol{q})$ denotes the quantity independent of parameters $\boldsymbol{\theta}$. Note that $P_F := F(F^\mathsf{T} F)^\dagger F^\mathsf{T}$ is the orthogonal projection on the subspace $\text{span}(F)$, so if $\text{span}(L) \subseteq \text{span}(F)$ we have $P_F L = L$, which finishes the proof. $\qquad\square$

**Lemma 10.** *In the case of a finite number $k$ of labels, for any task loss $L$, a surrogate loss $\Phi$ that is continuous and bounded from below, and a set of scores $\mathcal{F}$, the calibration function can be written as*

$$H_{\Phi,L,\mathcal{F}}(\varepsilon) = \min_{\substack{i,j \in \text{pred}(\mathcal{F}) \\ i \neq j}} H_{ij}(\varepsilon), \tag{20}$$

*where the set* $\text{pred}(\mathcal{F}) \subseteq \mathcal{Y}$ *is defined as the set of labels that the predictor can predict for some feasible scores and $H_{ij}$ is defined via minimization of the same objective as (5), but w.r.t. a smaller domain:*

$$H_{ij}(\varepsilon) = \inf_{\boldsymbol{f}, \boldsymbol{q}} \delta\phi(\boldsymbol{f}, \boldsymbol{q}), \tag{21}$$
$$\text{s.t. } \ell_i(\boldsymbol{q}) \leq \ell_j(\boldsymbol{q}) - \varepsilon,$$
$$\ell_i(\boldsymbol{q}) \leq \ell_c(\boldsymbol{q}), \quad \forall c \in \text{pred}(\mathcal{F}),$$
$$f_j \geq f_c, \quad \forall c \in \text{pred}(\mathcal{F}),$$
$$\boldsymbol{f} \in \mathcal{F},$$
$$\boldsymbol{q} \in \Delta_k.$$

*Here $\ell_c(\boldsymbol{q}) := (L\boldsymbol{q})_c$ is the expected loss if predicting label $c$. Index $i$ represents a label with the smallest expected loss while index $j$ represents a label with the largest score.*

*Proof.* We use the notation $\mathcal{F}_j$ to define the set of score vectors $\boldsymbol{f}$ where the predictor $\text{pred}(\boldsymbol{f})$ takes a value $j$, i.e., $\mathcal{F}_j := \{\boldsymbol{f} \in \mathcal{F} \mid \text{pred}(\boldsymbol{f}) = j\}$. The union of the sets $\mathcal{F}_j$, $j \in \text{pred}(\mathcal{F})$, equals the whole set $\mathcal{F}$. It is possible that sets $\mathcal{F}_j$ do not fully contain their boundary because of the usage of a particular tie-breaking strategy, but their closure can be expressed as $\overline{\mathcal{F}}_j := \{\boldsymbol{f} \in \mathcal{F} \mid f_j \geq f_c, \forall c \in \text{pred}(\mathcal{F})\}$.

If $\boldsymbol{f} \in \mathcal{F}_j$, i.e. $j = \text{pred}(\boldsymbol{f})$, then the feasible set of probability vectors $\boldsymbol{q}$ for which a label $i$ is one of the best possible predictions (i.e. $\delta\ell(\boldsymbol{f}, \boldsymbol{q}) = \ell_j(\boldsymbol{q}) - \ell_i(\boldsymbol{q}) \geq \varepsilon$) is

$$\Delta_{k,i,j,\varepsilon} := \{\boldsymbol{q} \in \Delta_k \mid \ell_i(\boldsymbol{q}) \leq \ell_c(\boldsymbol{q}), \forall c \in \text{pred}(\mathcal{F}); \ell_j(\boldsymbol{q}) - \ell_i(\boldsymbol{q}) \geq \varepsilon\},$$

because $\inf_{\boldsymbol{f}' \in \mathcal{F}} \ell(\boldsymbol{f}', \boldsymbol{q}) = \min_{c \in \text{pred}(\mathcal{F})} \ell_c(\boldsymbol{q})$.

The union of the sets $\{\mathcal{F}_j \times \Delta_{k,i,j,\varepsilon}\}_{i,j \in \text{pred}(\mathcal{F})}$ thus exactly equals the feasibility set of the optimization problem (5)-(6) (note that this is not true for the union of the sets $\{\overline{\mathcal{F}}_j \times \Delta_{k,i,j,\varepsilon}\}_{i,j \in \text{pred}(\mathcal{F})}$, which can be strictly larger), thus we can rewrite the definition of the calibration function as follows:

$$H_{\Phi,L,\mathcal{F}}(\varepsilon) = \min_{\substack{i,j \in \text{pred}(\mathcal{F}) \\ i \neq j}} \inf_{\substack{\boldsymbol{f} \in \mathcal{F}_j, \\ \boldsymbol{q} \in \Delta_{k,i,j,\varepsilon}}} \delta\phi(\boldsymbol{f}, \boldsymbol{q}). \tag{22}$$

To finish the proof, we use Lemma 27 of [49] claiming that the function $\delta\phi(\boldsymbol{f}, \boldsymbol{q})$ is continuous w.r.t. both $\boldsymbol{q}$ and $\boldsymbol{f}$, which allows us to substitute sets $\mathcal{F}_j$ in (22) with their closures $\overline{\mathcal{F}}_j$ without changing the value of the infimum. □

## C  Consistent surrogate losses

An ideal surrogate should not only be consistent, but also allow efficient optimization, by, e.g., being convex and allowing fast computation of stochastic gradients. In this paper, we study a generalization to arbitrary multi-class losses of a surrogate loss class from Zhang [49, Section 4.4.2][8] that satisfies these requirements:

$$\Phi_{a,b}(\boldsymbol{f}, \boldsymbol{y}) := \tfrac{1}{k} \sum\nolimits_{c=1}^{k} \big( L(c, \boldsymbol{y}) a(f_c) + b(f_c) \big), \tag{23}$$

where $a, b : \mathbb{R} \to \mathbb{R}$ are convex functions. A generic method to minimize this surrogate is to use any version of the SGD algorithm, while computing the stochastic gradient by sampling $\boldsymbol{y}$ from the data generating distribution and a label $c$ uniformly. In the case of the quadratic surrogate $\Phi_{\text{quad}}$, we proposed instead in the main paper to compute the sum over $c$ analytically instead of sampling $c$.

Extending the argument from Zhang [49], we show that the surrogates of the form (23) are consistent w.r.t. a task loss $L$ under some sufficient assumptions formalized in Theorem 11.

**Theorem 11** (Sufficient conditions for consistency). *The surrogate loss $\Phi_{a,b}$ is consistent w.r.t. a task loss $L$, i.e., $H_{\Phi_{a,b}, L, \mathbb{R}^k}(\varepsilon) > 0$ for any $\varepsilon > 0$, under the following conditions on the functions $a(f)$ and $b(f)$:*

  1. *The functions $a$ and $b$ are convex and differentiable.*
  2. *The function $ca(f) + b(f)$ is bounded from below and has a unique global minimizer (finite or infinite) for all $c \in [0, L_{max}]$.*
  3. *The functions $a(f)$ and $\frac{b'(f)}{a'(f)}$ are strictly increasing.*

*Proof.* Consider an arbitrary conditional probability vector $\boldsymbol{q} \in \Delta_k$. Assumption 2 then implies that the global minimizer $\boldsymbol{f}^*$ of the conditional surrogate risk $\phi(\boldsymbol{f}, \boldsymbol{q})$ w.r.t. $\boldsymbol{f}$ is unique. Assumption 1 allows us to set the derivatives to zero and obtain $\frac{b'(f_c^*)}{a'(f_c^*)} = -\ell_c(\boldsymbol{q})$ where $\ell_c(\boldsymbol{q}) := (L\boldsymbol{q})_c$. Assumption 3 then implies that $f_j^* \geq f_i^*$ holds if and only if $\ell_j(\boldsymbol{q}) \leq \ell_i(\boldsymbol{q})$.

Now, we will prove by contradiction that $H(\varepsilon) := H_{\Phi_{a,b}, L, \mathbb{R}^k}(\varepsilon) > 0$ for any $\varepsilon > 0$. Assume that for some $\varepsilon > 0$ we have $H(\varepsilon) = 0$. Lemma 10 then implies that for some $i, j \in \mathcal{Y}, i \neq j$, we have $H_{ij}(\varepsilon) = 0$. Note that the domain of (21) defining $H_{ij}$ is separable w.r.t. $\boldsymbol{q}$ and $\boldsymbol{f}$. We can now rewrite (21) as

$$H_{ij}(\varepsilon) = \inf_{\boldsymbol{q} \in \Delta_{k,i,j,\varepsilon}} \delta\phi^*(\boldsymbol{q}), \quad \text{where } \delta\phi^*(\boldsymbol{q}) := \inf_{\boldsymbol{f} \in \overline{\mathcal{F}}_j} \delta\phi(\boldsymbol{f}, \boldsymbol{q}),$$

where $\Delta_{k,i,j,\varepsilon}$ and $\overline{\mathcal{F}}_j$ are defined in the proof of Lemma 10. Lemma 27 of [49] implies that the function $\delta\phi^*(\boldsymbol{q})$ is a continuous function of $\boldsymbol{q}$. Given that $\Delta_{k,i,j,\varepsilon}$ is a compact set, the infimum is achieved at some point $\boldsymbol{q}^* \in \Delta_{k,i,j,\varepsilon}$. For this $\boldsymbol{q}^*$, the global minimum w.r.t. $\boldsymbol{f}$ exists (Assumption 2). The uniqueness of the global minimum implies that we have $f_j^* = \max_{c \in \mathcal{Y}} f_c^*$. The argument at the beginning of this proof then implies $\ell_j(\boldsymbol{q}^*) \leq \ell_i(\boldsymbol{q}^*)$ which contradicts the inequality $\ell_i(\boldsymbol{q}^*) \leq \ell_j(\boldsymbol{q}^*) - \varepsilon$ in the definition of $\Delta_{k,i,j,\varepsilon}$. □

Note that Theorem 11 actually proves that the surrogate $\Phi_{a,b}$ is order-preserving [49], which is a stronger property than consistency.

Below, we give several examples of possible functions $a(f)$, $b(f)$ that satisfy the conditions in Theorem 11 and their corresponding $f^*(\ell) := \operatorname{argmin}_{\boldsymbol{f} \in \mathbb{R}^k} \phi(\boldsymbol{f}, \boldsymbol{q})$ when $\ell := L\boldsymbol{q}$:

  1. If $a(f) = f$, $b(f) = \frac{f^2}{2}$ then $f^*(\ell) = -\ell$, leading to our quadratic surrogate (12).
  2. If $a(f) = \frac{1}{L_{\max}}(\exp(f) - \exp(-f))$, $b(f) = \exp(-f)$ then $f^*(\ell) = \frac{1}{2} \log(1 - \frac{1}{L_{\max}}\ell) - \frac{1}{2}\log(\frac{1}{L_{\max}}\ell)$.

3. If $a(f) = \frac{1}{L_{\max}} f$, $b(f) = \log(1 + \exp(-f))$ then $f^*(\ell) = \log(1 - \frac{1}{L_{\max}} \ell) - \log(\frac{1}{L_{\max}} \ell)$.

In the case of binary classification, these surrogates reduce to $L_2$-, exponential, and logistic losses, respectively.

## D  Bounds on the calibration function

### D.1  Lower bound

**Theorem 7** (Lower bound on $H_{\Phi_{\text{quad}}}$). *For any task loss $L$, its quadratic surrogate $\Phi_{\text{quad}}$, and a score subspace $\mathcal{F}$ containing the column space of $L$, the calibration function can be lower bounded:*

$$H_{\Phi_{\text{quad}}, L, \mathcal{F}}(\varepsilon) \geq \frac{\varepsilon^2}{2k \max_{i \neq j} \|P_{\mathcal{F}} \Delta_{ij}\|_2^2} \geq \frac{\varepsilon^2}{4k},$$

*where $P_{\mathcal{F}}$ is the orthogonal projection on the subspace $\mathcal{F}$ and $\Delta_{ij} = \mathbf{e}_i - \mathbf{e}_j \in \mathbb{R}^k$ with $\mathbf{e}_c$ being the $c$-th basis vector of the standard basis in $\mathbb{R}^k$.*

*Proof.* First, let us assume that the score subspace $\mathcal{F}$ is defined as the column space of a matrix $F \in \mathbb{R}^{k \times r}$, i.e., $\boldsymbol{f}(\boldsymbol{\theta}) = F\boldsymbol{\theta}$. Lemma 9 gives us expression (13) for $\delta\phi_{\text{quad}}(F\boldsymbol{\theta}, \boldsymbol{q})$, which is jointly convex w.r.t. a conditional probability vector $\boldsymbol{q}$ and parameters $\boldsymbol{\theta}$.

The optimization problem (5)-(6) is non-convex because the constraint (6) on the excess risk depends of the predictor function pred($\boldsymbol{f}$), see Eq. (1), containing the argmax operation. However, if we constrain the predictor to output label $j$, i.e., $f_j \geq f_c$, $\forall c$, and the label delivering the smallest possible expected loss to be $i$, i.e., $(L\boldsymbol{q})_i \leq (L\boldsymbol{q})_c$, $\forall c$, the problem becomes convex because all the constraints are linear and the objective is convex. Lemma 10 in App. B allows to bound the calibration function with the minimization w.r.t. selected labels $i$ and $j$, $H_{\Phi_{\text{quad}}, L, \mathcal{F}}(\varepsilon) \geq \min_{i \neq j} H_{ij}(\varepsilon)$,[9] where $H_{ij}(\varepsilon)$ is defined as follows:

$$H_{ij}(\varepsilon) = \min_{\boldsymbol{\theta}, \boldsymbol{q}} \frac{1}{2k} \|F\boldsymbol{\theta} + L\boldsymbol{q}\|_2^2, \tag{24}$$

$$\text{s.t. } (L\boldsymbol{q})_i \leq (L\boldsymbol{q})_j - \varepsilon,$$
$$(L\boldsymbol{q})_i \leq (L\boldsymbol{q})_c, \ \forall c \in \text{pred}(\mathcal{F})$$
$$(F\boldsymbol{\theta})_j \geq (F\boldsymbol{\theta})_c, \ \forall c \in \text{pred}(\mathcal{F})$$
$$\boldsymbol{q} \in \Delta_k.$$

To obtain a lower bound, we relax (24) by removing some of the constraints and arrive at

$$H_{ij}(\varepsilon) \geq \min_{\boldsymbol{\theta}, \boldsymbol{q}} \frac{1}{2k} \|F\boldsymbol{\theta} + L\boldsymbol{q}\|_2^2, \tag{25}$$

$$\text{s.t. } \Delta_{ij}^{\mathsf{T}} L\boldsymbol{q} \leq -\varepsilon, \tag{26}$$

$$\Delta_{ij}^{\mathsf{T}} F\boldsymbol{\theta} \leq 0, \tag{27}$$

where $\Delta_{ij}^{\mathsf{T}} L\boldsymbol{q} = (L\boldsymbol{q})_i - (L\boldsymbol{q})_j$, $\Delta_{ij}^{\mathsf{T}} F\boldsymbol{\theta} = (F\boldsymbol{\theta})_i - (F\boldsymbol{\theta})_j$, and $\Delta_{ij} = \mathbf{e}_i - \mathbf{e}_j \in \mathbb{R}^k$ with $\mathbf{e}_c \in \mathbb{R}^k$ being a vector of all zeros with 1 at position $c$.

The constraint (26) can be readily substituted with equality

$$\Delta_{ij}^{\mathsf{T}} L\boldsymbol{q} = -\varepsilon, \tag{28}$$

without changing the minimum because multiplication of both $\boldsymbol{q}$ and $\boldsymbol{\theta}$ by the constant $\frac{-\varepsilon}{\Delta_{ij}^{\mathsf{T}} L\boldsymbol{q}} \in (0, 1]$ preserves feasibility and can only decrease the objective (25).

We now explicitly solve the resulting constraint optimization problem via the KKT optimality conditions. The stationarity constraints give us

$$\frac{1}{k} F^{\mathsf{T}}(F\boldsymbol{\theta} + L\boldsymbol{q}) + \mu F^{\mathsf{T}} \Delta_{ij} = 0, \tag{29}$$

$$\frac{1}{k} L^{\mathsf{T}}(F\boldsymbol{\theta} + L\boldsymbol{q}) + \nu L^{\mathsf{T}} \Delta_{ij} = 0; \tag{30}$$

the complementary slackness gives $\mu \Delta_{ij}^\mathsf{T} F\boldsymbol{\theta} = 0$ and the feasibility constraints give (28), (27), and $\mu \geq 0$.

Equation (29) allows to compute

$$\boldsymbol{\theta} = -(F^\mathsf{T} F)^\dagger (k\mu F^\mathsf{T} \Delta_{ij} + F^\mathsf{T} L\boldsymbol{q}). \tag{31}$$

By substituting (31) into (30) and by using the identity (because $L \in \mathrm{span}(F)$):

$$P_F L = F(F^\mathsf{T} F)^\dagger F^\mathsf{T} L = L, \tag{32}$$

we get $(\mu - \nu) L^\mathsf{T} \Delta_{ij} = 0$. If $L^\mathsf{T} \Delta_{ij} = 0$, the problem (25), (27), (28) is infeasible for $\varepsilon > 0$ implying $H_{ij}(\varepsilon) = +\infty$. Otherwise, we have $\mu = \nu$.

From (31) and (32), we also have that:

$$F\boldsymbol{\theta} + L\boldsymbol{q} = -k\mu P_F \Delta_{ij}. \tag{33}$$

By plugging (31) into the complementary slackness condition and combining with (28), we get

$$\mu^2 k \|P_F \Delta_{ij}\|_2^2 = \mu\varepsilon$$

implying that either $\mu = 0$ or $\mu k \|P_F \Delta_{ij}\|_2^2 = \varepsilon$. In the first case, Eq. (33) implies $F\boldsymbol{\theta} = -L\boldsymbol{q}$ making satisfying both (28) and (27) impossible. Thus, the later is satisfied implying that the objective (25) is equal to[10]

$$\tfrac{1}{2k}\|F\boldsymbol{\theta} + L\boldsymbol{q}\|_2^2 = \tfrac{\varepsilon^2}{2k\|P_F \Delta_{ij}\|_2^2}.$$

Finally, orthogonal projections contract the $L_2$-norm, thus $\|P_F \Delta_{ij}\|_2^2 \leq 2$, which gives the second lower bound in the statement of the theorem and finishes the proof. $\qquad\square$

## D.2 Upper bound

**Theorem 8** (Upper bound on $H_{\Phi_\text{quad}}$). *If a loss matrix $L$ with $L_{max} > 0$ defines a pseudometric[7] on labels and there are no constraints on the scores, i.e., $\mathcal{F} = \mathbb{R}^k$, then the calibration function for the quadratic surrogate $\Phi_\text{quad}$ can be upper bounded:*

$$H_{\Phi_\text{quad}, L, \mathcal{F}}(\varepsilon) \leq \tfrac{\varepsilon^2}{2k}, \quad 0 \leq \varepsilon \leq L_{max}.$$

*Proof.* After applying Lemmas 9 and 10, we arrive at

$$H_{ij}(\varepsilon) = \inf_{\boldsymbol{f}, \boldsymbol{q}} \tfrac{1}{2k}\|\boldsymbol{f} + L\boldsymbol{q}\|^2, \tag{34}$$

$$\text{s.t. } \ell_i(\boldsymbol{q}) \leq \ell_j(\boldsymbol{q}) - \varepsilon,$$
$$\ell_i(\boldsymbol{q}) \leq \ell_c(\boldsymbol{q}), \quad \forall c \in \mathcal{Y},$$
$$f_j \geq f_c, \quad \forall c \in \mathcal{Y},$$
$$\boldsymbol{f} \in \mathbb{R}^k, \quad \boldsymbol{q} \in \Delta_k.$$

We now consider labels $i$ and $j$ such that $L_{ij} = L_\text{max} > 0$ and the point $q_i = \tfrac{1}{2} + \tfrac{\varepsilon}{2L_{ij}}$, $q_j = \tfrac{1}{2} - \tfrac{\varepsilon}{2L_{ij}}$ (non-negative for $\varepsilon \leq L_\text{max}$). We let $q_c = 0$ for $c \notin \{i, j\}$, $f_j = f_i = -\ell_i(\boldsymbol{q})$ and $f_c = -\ell_c(\boldsymbol{q})$ for $c \notin \{i, j\}$. We now show that this assignment is feasible.

We have $\ell_j(\boldsymbol{q}) = q_i L_{ji} + q_j L_{jj} = q_i L_{ji} = q_i L_{ij}$ by symmetry of $L$. Similarly, $\ell_i(\boldsymbol{q}) = q_j L_{ij}$ and thus

$$\ell_j(\boldsymbol{q}) - \ell_i(\boldsymbol{q}) = L_{ij}\tfrac{\varepsilon}{L_{ij}} = \varepsilon.$$

We also have

$$\ell_c(\boldsymbol{q}) - \ell_i(\boldsymbol{q}) = q_i L_{ci} + q_j L_{cj} - q_j L_{ij} \geq q_j(L_{ic} + L_{cj} - L_{ij}) \geq 0.$$

The first inequality uses $q_i \geq q_j$ and the second inequality uses the fact that $L$ satisfies the triangle inequality (as a pseudometric). Finally, $f_j - f_c = -\ell_i(\boldsymbol{q}) + \ell_c(\boldsymbol{q}) \geq 0$.

We thus have shown that the defined point is feasible, so we compute its objective value. We have

$$\tfrac{1}{2k}\|\boldsymbol{f} + L\boldsymbol{q}\|^2 = \tfrac{1}{2k}(\ell_j(\boldsymbol{q}) - \ell_i(\boldsymbol{q}))^2 = \tfrac{\varepsilon^2}{2k},$$

which completes the proof. $\qquad\square$

### D.3 Computation of the lower bounds for specific task losses

**0-1 loss.** Let $L_{01}$ denote the loss matrix of the 0-1 loss, i.e., $L_{01}(i,j) := [i \neq j]$.[2] It is convenient to rewrite it with a matrix notation $L_{01} = \mathbf{1}_k \mathbf{1}_k^\top - \mathbf{I}_k$, where $\mathbf{1}_k \in \mathbb{R}^k$ is the vector of all ones and $\mathbf{I}_k \in \mathbb{R}^{k \times k}$ is the identity matrix. We have $\mathrm{rank}(L_{01}) = k$ (for $k \geq 2$), thus $\mathrm{span}(L) = \mathbb{R}^k$. By putting no constraints on the scores, we can easily apply Theorem 7 and obtain the lower bound of $\frac{\varepsilon^2}{4k}$, which is shown to be tight in Proposition 12 of Section E.1.

**Block 0-1 loss.** We use the symbol $L_{01,b}$ to denote the loss matrix of the block 0-1 loss with $b$ blocks, i.e., $L_{01,b}(i,j) := [i \text{ and } j \text{ are not in the same block}]$. We use $s_v$ to denote the size of block $v$, $v = 1, ..., b$, and then $s_1 + \cdots + s_b = k$. In the case when all the blocks are of equal sizes, we denote their size by $s$ and have $k = bs$.

With a matrix notation, we have $L_{01,b} = \mathbf{1}_k \mathbf{1}_k^\top - UU^\top$ where the columns of the matrix $U \in \mathbb{R}^{k \times b}$ are indicators of the blocks. We have $\mathrm{rank}(L_{01,b}) = b$ and can simply define $\mathcal{F}_{01,b} := \mathrm{span}(F_{01,b})$ with $F_{01,b} := U$. If we assume that all the blocks have equal size, then we have $U^\top U = s\mathbf{I}_b$ and $\|P_{\mathcal{F}_{01,b}} \Delta_{ij}\|_2^2 = \frac{2}{s}$ if labels $i$ and $j$ belong to different blocks, while $P_{\mathcal{F}_{01,b}} \Delta_{ij} = 0$ if $i$ and $j$ belong to the same block. This leads to the lower bound $\frac{\varepsilon^2}{4b}$, which is shown to be tight in Proposition 14 of Section E.2.

**Hamming loss.** Consider the (normalized) Hamming loss between tuples of $T$ binary variables, where $\hat{y}_t$ and $y_t$ are the $t$-th variables of a prediction $\hat{\boldsymbol{y}}$ and a correct label $\boldsymbol{y}$, respectively:

$$
\begin{aligned}
L_{\mathrm{Ham},T}(\hat{\boldsymbol{y}}, \boldsymbol{y}) &:= \frac{1}{T} \sum_{t=1}^{T} [\hat{y}_t \neq y_t] \qquad (35) \\
&= \frac{1}{T} \sum_{t=1}^{T} ([\hat{y}_t = 0][y_t = 1] + [\hat{y}_t = 1][y_t = 0]) \\
&= \frac{1}{T} \sum_{t=1}^{T} (1 - [\hat{y}_t = 1])[y_t = 1] + [\hat{y}_t = 1][y_t = 0]) \\
&= \frac{1}{T} \sum_{t=1}^{T} [y_t = 1] + \frac{1}{T} \sum_{t=1}^{T} ([y_t = 0] - [y_t = 1]) \ [\hat{y}_t = 1] \\
&= \alpha_0(\boldsymbol{y}) + \sum_{t=1}^{T} \alpha_t(\boldsymbol{y})[\hat{y}_t = 1],
\end{aligned}
$$

The vectors $\boldsymbol{\alpha}_t(\cdot)$ depend only on the column index of the loss matrix. The decomposition (35) implies that $\mathcal{F}_{\mathrm{Ham},T} := \mathrm{span}(F_{\mathrm{Ham},T})$ equals to $\mathrm{span}(L_{\mathrm{Ham},T})$ for $F_{\mathrm{Ham},T} := [\frac{1}{2}\mathbf{1}_{2^T}, \boldsymbol{h}^{(1)}, \ldots, \boldsymbol{h}^{(T)}]$, $(\boldsymbol{h}^{(t)})_{\hat{\boldsymbol{y}}} := [\hat{y}_t = 1]$, $t = 1, \ldots, T$. We also have that $\mathrm{rank}(L_{\mathrm{Ham},T}) = \mathrm{rank}(F_{\mathrm{Ham},T}) = T + 1$.

In Section G, we show that $\max_{i \neq j} \|P_{\mathcal{F}_{\mathrm{Ham},T}} \Delta_{ij}\|_2^2 = \frac{4T}{2^T}$. By plugging this identity into the lower bound (14), we get $H_{\Phi_{\mathrm{quad}}, L_{\mathrm{Ham},T}, \mathcal{F}_{\mathrm{Ham},T}} \geq \frac{\varepsilon^2}{8T}$, which appears to be tight according to Proposition 15 of Section E.3.

**Non-tight cases.** In the cases of the block 0-1 loss and the mixed 0-1 and block 0-1 loss (Propositions 13 and 16, respectively), we observe gaps between the lower bound (14) and the exact calibration functions, which shows the limitations of the bound. In particular, it cannot detect level-$\eta$ consistency for $\eta > 0$ (see Definition 3) and does not change when the loss changes, but the score subspace stays the same.

## E Exact calibration functions for quadratic surrogate

This section presents our derivations for the exact values of the calibration functions for different losses. While doing these derivations, we have used numerical simulations and symbolic derivations to check for correctness. Our numerical and symbolic tools are available online.[11]

### E.1 0-1 loss

**Proposition 12.** *Let $L_{01}$ be the 0-1 loss, i.e., $L_{01}(i,j) = [i \neq j]$. Then, the calibration function equals the following quadratic function w.r.t. $\varepsilon$:*

$$
H_{\Phi_{\mathrm{quad}}, L_{01}, \mathbb{R}^k}(\varepsilon) = \frac{\varepsilon^2}{4k}, \quad 0 \leq \varepsilon \leq 1. \qquad (36)
$$

Note that in the case of binary classification, the function (36) is equal to the calibration function for the least squares and truncated least squares surrogates [5, 43].

*Proof.* First, Lemma 9 with $\mathcal{F} = \mathbb{R}^k$ and $F = \mathbf{I}_k$ gives us the expression

$$\delta\phi_{\text{quad}}(F\boldsymbol{\theta}, \boldsymbol{q}) = \tfrac{1}{2k}\|\boldsymbol{f} + L\boldsymbol{q}\|_2^2, \tag{37}$$

with $\boldsymbol{f} = \boldsymbol{\theta} \in \mathbb{R}^k$.

We now reduce the optimization problem (5)-(6) to a convex one by using Lemma 10 and by writing $H_{\Phi_{\text{quad}}, L_{01}, \mathbb{R}^k}(\varepsilon) = \min_{i \neq j \in \mathcal{Y}} H_{ij}(\varepsilon)$, which holds because $\text{pred}(\mathbb{R}^k) = \mathcal{Y}$. Because of the symmetries of the 0-1 loss, all the choices of $i$ and $j$ give the same (up to a permutation of labels) optimization problem to compute $H_{ij}(\varepsilon)$. The definition of the 0-1 loss implies $(L\boldsymbol{q})_c = 1 - q_c$, which simplifies the excess of the expected task loss appearing in (6) to $\delta\ell(\boldsymbol{f}, \boldsymbol{q}) = (L\boldsymbol{q})_j - (L\boldsymbol{q})_i = q_i - q_j$. After putting all these together, we get

$$H_{ij}(\varepsilon) = \min_{\boldsymbol{f}, \boldsymbol{q}} \tfrac{1}{2k} \sum_{c=1}^{k} (f_c + 1 - q_c)^2, \tag{38}$$

$$\text{s.t. } q_i \geq q_j + \varepsilon,$$
$$q_i \geq q_c, \quad c = 1, \ldots, k,$$
$$f_j \geq f_c, \quad c = 1, \ldots, k,$$
$$\sum_{c=1}^{k} q_c = 1, \quad q_c \geq 0.$$

We claim that there exists an optimal point of (38), $\boldsymbol{f}^*$, $\boldsymbol{q}^*$, such that $q_c^* = 0$, $c \notin \{i, j\}$, $q_i^* = \tfrac{1}{2} + \tfrac{\varepsilon}{2}$, $q_j^* = \tfrac{1}{2} - \tfrac{\varepsilon}{2}$; $f_c^* = -1$, $c \notin \{i, j\}$, $f^* := f_i^* = f_j^*$. Note that apart from the specific value of $f^*$, this is the same point used to prove the upper bound of Theorem 8. After proving this, we will minimize the objective w.r.t. remaining scores at this point.[12]

First, if any $q_c^* = \delta > 0$, $c \notin \{i, j\}$, we can safely move this probability mass to $q_i$ and $q_j$ with the operation

$$q_c^* := q_c^* - \delta = 0, \qquad q_i^* := q_i^* + \tfrac{\delta}{2}, \qquad q_j^* := q_j^* + \tfrac{\delta}{2},$$
$$f_c^* := f_c^* - \delta, \qquad f_i^* := f_i^* + \tfrac{\delta}{2}, \qquad f_j^* := f_j^* + \tfrac{\delta}{2},$$

which keeps all the constraints of (38) feasible and does not change the objective value.

Second, all the scores $f_c^*$ have to belong to the segment $[-1, 0]$ otherwise clipping them will decrease the objective. With this, setting $f_c^* := -1$, $c \notin \{i, j\}$ can only decrease the objective and will not violate the constraints.

We now show that the equality $q_i^* = q_j^* + \varepsilon$ can hold at the optimum. Indeed, if $q_i^* - q_j^* = \delta' > \varepsilon$, the operation

$$q_i^* := q_i^* - \tfrac{\delta' - \varepsilon}{2}, \qquad\qquad q_j^* := q_j^* + \tfrac{\delta' - \varepsilon}{2}, \tag{39}$$
$$f_i^* := f_i^* - \tfrac{\delta' - \varepsilon}{2}, \qquad\qquad f_j^* := f_j^* + \tfrac{\delta' - \varepsilon}{2}.$$

keeps the objective the same and maintains the feasibility constraints. So combining with $q_i^* + q_j^* = 1$, we can now conclude that $q_i^* = \tfrac{1}{2} + \tfrac{\varepsilon}{2}$, $q_j^* = \tfrac{1}{2} - \tfrac{\varepsilon}{2}$ is an optimal point.

We now show that the equality $f_i^* = f_j^*$ can hold at the optimum. First, we know that the values $f_i^*$ and $f_j^*$ belong to the segment $[q_j^* - 1, q_i^* - 1]$, otherwise we can always truncate the values to the borders of the segment and get an improvement of the objective. Finally, since the inequality $f_j^* \geq f_i^*$ must hold, we conclude that $f_i^* = f_j^* := f^*$ so that $f_i^*$ is closest to its target $q_i^* - 1$ to minimize the objective.

At the optimal point defined above, it remains to find the value $f^*$ delivering the minimum of the objective. We can achieve this by computing

$$H_{ij}(\varepsilon) = \tfrac{1}{2k} \min_{f \in [-1,0]} (f + \tfrac{1}{2} - \tfrac{\varepsilon}{2})^2 + (f + \tfrac{1}{2} + \tfrac{\varepsilon}{2})^2,$$

which implies $f^* = -0.5$ and $H_{\Phi_{\text{quad}}, L_{01}, \mathbb{R}^k}(\varepsilon) = \tfrac{\varepsilon^2}{4k}$. $\qquad\square$

**Remark.** We note that the conditional distribution used in the proof above, $q_i = \tfrac{1}{2} + \tfrac{\varepsilon}{2}$, $q_j = \tfrac{1}{2} - \tfrac{\varepsilon}{2}$, $q_c = 0$, $c \notin \{i, j\}$, is somewhat unsatisfying from the perspective of explaining why learning the 0-1 loss might be difficult. Indeed, it looks like a gradient based learning algorithm that would start with all values $f_c = -1$ would at the end only optimize over $f_i$ and $f_j$ as the gradient with respect to $f_c$ for $c \notin \{i, j\}$ would stay at zero in $\Phi_{\text{quad}}(\boldsymbol{f}, \boldsymbol{y})$ (12) given that only $i$ or $j$ could appear in $\boldsymbol{y}$. From this observation, one could think that the calibration function perspective is misleading as SGD could have faster convergence rate than predicted by the worst case for this situation. Fortunately, one can easily check that the point $q_i = \tfrac{1}{3} + \tfrac{\varepsilon}{2}$, $q_j = \tfrac{1}{3} - \tfrac{\varepsilon}{2}$, $q_c = \tfrac{1}{3(k-2)}$ for $c \notin \{i, j\}$, $f_i = f_j = \tfrac{1}{3}$ and $f_c = -\ell_c(\boldsymbol{q})$ for $c \notin \{i, j\}$ is feasible for (38) and yields the same optimal value of $\tfrac{\varepsilon^2}{4k}$ for the objective, thus providing another example where the exponential multiclass nature is more readily apparent and cannot be fixed by some "natural initialization" of the learning algorithm.

## E.2 Block 0-1 loss

Recall that $L_{01,b}$ is the block 0-1 loss, i.e., $L_{01,b}(i, j) = [i$ and $j$ are not in the same block$]$. We use $b$ to denote the total number of blocks and $s_v$ to denote the size of block $v$, $v = 1, ..., b$. In this section, we compute the calibration functions for the case of unconstrained scores (Proposition 13) and for the case of the scores belonging to the column span of the loss matrix (Proposition 14).

**Proposition 13.** *Without constraints on the scores, the calibration function for the block 0-1 loss equals the following quadratic function w.r.t. $\varepsilon$:*

$$H_{\Phi_{\text{quad}}, L_{01,b}, \mathbb{R}^k}(\varepsilon) = \tfrac{\varepsilon^2}{4k} \min_{v=1,\ldots,b} \tfrac{2s_v}{s_v + 1} \leq \tfrac{\varepsilon^2}{2k}, \quad 0 \leq \varepsilon \leq 1.$$

Note that when $s_v = 1$ for some $v$, we have $H_{\Phi_{\text{quad}}, L_{01,b}, \mathbb{R}^k}(\varepsilon)$ matching to the $\tfrac{\varepsilon^2}{4k}$ lower bound of Theorem 7. When $s_v \to \infty$ for all blocks, we have $H_{\Phi_{\text{quad}}, L_{01,b}, \mathbb{R}^k}(\varepsilon)$ matching to the $\tfrac{\varepsilon^2}{2k}$ upper bound of Theorem 8.

*Proof.* This proof is of the same structure as the proof of Proposition 12 above.

We use $b(i) \in 1, \ldots, b$ to denote the block to which label $i$ belongs and $\mathcal{Y}_v$ to denote the set of labels that belong to block $v$. We also use $Q_v$, $v \in 1, \ldots, b$, as a shortcut to $\sum_{i \in \mathcal{Y}_v} q_i$, which is the total probability mass on block $v$.

We start by noting that the $i$-th component of the vector $(L_{01,b})\boldsymbol{q}$ equals $1 - Q_{b(i)}$. By applying Lemmas 9, 10, we get

$$H_{ij}(\varepsilon) = \min_{\boldsymbol{f}, \boldsymbol{q}} \quad \tfrac{1}{2k} \sum_{v=1}^{b} \sum_{c \in \mathcal{Y}_v} (f_c + 1 - Q_{b(c)})^2, \tag{40}$$

$$Q_{b(i)} - Q_{b(j)} \geq \varepsilon, \tag{41}$$
$$Q_{b(i)} \geq Q_u, \ u = 1, \ldots, b,$$
$$f_j \geq f_c, \ c = 1, \ldots, k,$$
$$\sum_{c=1}^{k} q_c = 1, \quad q_c \geq 0.$$

Analogously to Proposition 12, we claim that there exists an optimal point of (40) such that $q_c = 0$, $c \notin \{i, j\}$; $q_i = 0.5 + \tfrac{\varepsilon}{2} = Q_{b(i)}$; $q_j = 0.5 - \tfrac{\varepsilon}{2} = Q_{b(j)}$; $f_c = -1$, $c \notin \mathcal{Y}_{ij} := \mathcal{Y}_{b(i)} \cup \mathcal{Y}_{b(j)}$.

At first, note that if $b(i) = b(j)$, then the constraint (41) is never feasible, so we'll assume that $b(i) \neq b(j)$.

We will now show that we can consider only configurations with all the probability mass on the two selected blocks. Consider some optimal point $\boldsymbol{f}^*$, $\boldsymbol{q}^*$ and denote with $\delta = \sum_{c \in \mathcal{Y} \setminus \mathcal{Y}_{ij}} q_c^*$ the probability mass on the unselected blocks. The operation

$$f_c^* := f_c^* + \tfrac{\delta}{2}, \ c \in \mathcal{Y}_{ij}, \qquad\qquad f_c^* := -1, \ c \notin \mathcal{Y}_{ij}$$
$$q_i^* := q_i^* + \tfrac{\delta}{2}, q_j^* := q_j^* + \tfrac{\delta}{2}, \qquad\qquad q_c^* := 0, \ c \notin \mathcal{Y}_{ij}$$

can only decrease the objective of (40) because the summands corresponding to the unselected blocks are set to zero. All the constraints stay feasible and the summands corresponding to the selected blocks keep their values.

The probability mass within the block $b(i)$ can be safely moved to $q_i^*$ without changing the objective or violating any constraints. Analogously, the probability mass within the block $b(j)$ can be safely moved to $q_j^*$. By reusing the operation (39), we can now ensure that $q_i^* = q_j^* + \varepsilon$ and thus that $q_i^* = \tfrac{1}{2} + \tfrac{\varepsilon}{2}$ and $q_j^* = \tfrac{1}{2} - \tfrac{\varepsilon}{2}$.

At the point defined above, we now minimize the objective (40) w.r.t. $f_c$, $c \in \mathcal{Y}_{ij}$. At an optimal point, all values $f_c^*$, $c \in \mathcal{Y}_{ij}$, belong to the segment $[Q_{b(j)}^* - 1, Q_{b(i)}^* - 1]$, otherwise we can always truncate the values to the borders of the segment and get an improvement of the objective. For all the scores $f_c^*$, $c \neq j$, the following identity holds

$$f_c^* = \begin{cases} Q_{b(c)}^* - 1, \text{ if } Q_{b(c)}^* - 1 < f_j^*, \\ f_j^*. \end{cases} \tag{42}$$

Combining with the segment constraint, it implies that in the block of the label $i$, we have $f_c^* = f_j^*$, $c \in \mathcal{Y}_{b(i)}$, and, in the block of the label $j$, we have $f_c^* = Q_{b(j)}^* - 1, c \in \mathcal{Y}_{b(j)} \setminus j$.

By plugging the obtained values of $q_c^*$ and $f_c^*$ into (40) and denoting the value $f_j^* + 0.5$ with $\tilde{f}$, we get

$$H_{ij}(\varepsilon) = \min_{\tilde{f}} \tfrac{1}{2k} \left( s_{b(i)}(\tilde{f} - \tfrac{\varepsilon}{2})^2 + (\tilde{f} + \tfrac{\varepsilon}{2})^2 \right), \tag{43}$$
$$\text{s.t. } \tilde{f} \in [-\tfrac{\varepsilon}{2}, \tfrac{\varepsilon}{2}].$$

By setting the derivative of the objective (43) to zero, we get

$$\tilde{f} = \tfrac{\varepsilon}{2} \tfrac{s_{b(i)} - 1}{s_{b(i)} + 1},$$

which belongs to the segment $[-\tfrac{\varepsilon}{2}, \tfrac{\varepsilon}{2}]$. We compute the function value at this point:

$$H_{ij}(\varepsilon) = \tfrac{\varepsilon^2}{4k} \tfrac{2 s_{b(i)}}{s_{b(i)} + 1},$$

which finishes the proof. $\qquad\qquad\qquad\qquad\qquad\qquad\qquad\qquad\qquad\qquad\qquad\qquad\qquad\quad \square$

**Proposition 14.** *Let the scores $\boldsymbol{f}$ be piecewise constant on the blocks of the loss, i.e. belong to the subspace $\mathcal{F}_{01,b} = \operatorname{span}(L_{01,b}) \subseteq \mathbb{R}^k$. Then, the calibration function equals the following quadratic function w.r.t. $\varepsilon$:*

$$H_{\Phi_{\mathrm{quad}}, L_{01,b}, \mathcal{F}_{01,b}}(\varepsilon) = \tfrac{\varepsilon^2}{4k} \min_{v \neq u} \tfrac{2 s_v s_u}{s_v + s_u}, \quad 0 \leq \varepsilon \leq 1.$$

*If all the blocks are of the same size, we have $H_{\Phi_{\mathrm{quad}}, L_{01,b}, \mathcal{F}_{01,b}}(\varepsilon) = \tfrac{\varepsilon^2}{4b}$ where $b$ is the number of blocks.*

*Proof.* The constraints on scores $\boldsymbol{f} \in \mathcal{F}_{01,b}$ simply imply that the scores within all the blocks are equal. Having this in mind, the proof exactly matches the proof of Proposition 13 until the argument around Eq. (42). Now we cannot set the scores of the block $b(j)$ to different values, and, thus they are all equal to $f^*$.

By plugging the obtained values of $q_c^*$ and $f_c^*$ into (40) and denoting the value $f_j^* + 0.5$ with $\tilde{f}$, we get

$$H_{ij}(\varepsilon) = \min_{\tilde{f}} \tfrac{1}{2k} \left( s_{b(i)}(\tilde{f} - \tfrac{\varepsilon}{2})^2 + s_{b(j)}(\tilde{f} + \tfrac{\varepsilon}{2})^2 \right),$$
$$\text{s.t. } \tilde{f} \in [-\tfrac{\varepsilon}{2}, \tfrac{\varepsilon}{2}]. \tag{44}$$

By setting the derivative of the objective (44) to zero, we get

$$\tilde{f} = \frac{\varepsilon}{2}\frac{s_{b(i)}-s_{b(j)}}{s_{b(i)}+s_{b(j)}},$$

which belongs to the segment $[-\frac{\varepsilon}{2}, \frac{\varepsilon}{2}]$. We now compute the function value at this point:

$$H_{ij}(\varepsilon) = \frac{\varepsilon^2}{4k}\frac{2s_{b(i)}s_{b(j)}}{s_{b(i)}+s_{b(j)}},$$

which finishes the proof. □

### E.3 Hamming loss

Recall that $L_{\text{Ham},T}$ is the Hamming loss defined over $T$ binary variables (see Eq. (35) for the precise definition). In this section, we compute the calibration function for the case of the scores belonging to the column span of the loss matrix (Proposition 15).

**Proposition 15.** *Assume that the scores $\boldsymbol{f}$ always belong to the column span of the Hamming loss matrix $L_{\text{Ham},T}$, i.e., $\mathcal{F}_{\text{Ham},T} = \text{span}(L_{\text{Ham},T}) \subseteq \mathbb{R}^k$. Then, the calibration function can be computed as follows:*

$$H_{\Phi_{\text{quad}}, L_{\text{Ham},T}, \mathcal{F}_{\text{Ham},T}}(\varepsilon) = \frac{\varepsilon^2}{8T}, \quad 0 \le \varepsilon \le 1.$$

*Proof.* We start the proof by applying Lemma 10 and by studying the vector of the expected losses $(L_{\text{Ham},T})\boldsymbol{q}$. We note that the $\hat{\boldsymbol{y}}$-th element $\ell_{\hat{\boldsymbol{y}}}(\boldsymbol{q})$, $\hat{\boldsymbol{y}} = (\hat{y}_t)_{t=1}^T$, $\hat{y}_t \in \{0,1\}$, has a simple form of

$$\ell_{\hat{\boldsymbol{y}}}(\boldsymbol{q}) = \sum_{\boldsymbol{y}\in\mathcal{Y}} \frac{q_{\boldsymbol{y}}}{T}\sum_{t=1}^T[\hat{y}_t \ne y_t] = 1 - \frac{1}{T}\sum_{t=1}^T\sum_{\boldsymbol{y}\in\mathcal{Y}} q_{\boldsymbol{y}}[\hat{y}_t = y_t].$$

The quantity $\sum_{\boldsymbol{y}\in\mathcal{Y}} q_{\boldsymbol{y}}[\hat{y}_t = y_t]$ corresponds to the marginal probability of a variable $t$ taking a label $\hat{y}_t$. Note that the expected loss $\ell_{\hat{\boldsymbol{y}}}(\boldsymbol{q})$ only depends on $\boldsymbol{q}$ through marginal probabilities, thus two distributions $\boldsymbol{q}_1$ and $\boldsymbol{q}_2$ with the same marginals would be indistinguishable when plugged in the optimization problem for $H_{ij}(\varepsilon)$ (21), given that both the constraints and the objective (by Lemma 9) only depend on $\boldsymbol{q}$ through the expected loss $\ell_{\hat{\boldsymbol{y}}}(\boldsymbol{q})$. Having this in mind, we can consider only separable distributions, i.e., $q_{\boldsymbol{y}} = \prod_{t=1}^T\big(q_t[y_t = 1] + (1-q_t)[y_t = 0]\big)$, where $q_t \in [0,1]$, $t = 1, \ldots, T$, are the parameters defining the distribution.

By combining the notation above with Lemmas 9 and 10, we arrive at the following optimization problem:

$$H_{\tilde{\boldsymbol{y}}\hat{\boldsymbol{y}}}(\varepsilon) = \min_{\boldsymbol{f},\boldsymbol{q}} \frac{1}{2k}\sum_{\boldsymbol{y}\in\mathcal{Y}}\left(f_{\boldsymbol{y}}+1-\frac{1}{T}\sum_{t=1}^T q_{t,y_t}\right)^2, \tag{45}$$

$$\text{s.t. } \frac{1}{T}\sum_{t=1}^T (q_{t,\tilde{y}_t}-q_{t,\hat{y}_t}) \ge \varepsilon, \tag{46}$$

$$\frac{1}{T}\sum_{t=1}^T (q_{t,\tilde{y}_t}-q_{t,y_t}) \ge 0, \ \forall\boldsymbol{y}\in\mathcal{Y}, \tag{47}$$

$$f_{\hat{\boldsymbol{y}}} \ge f_{\boldsymbol{y}}, \ \forall\boldsymbol{y}\in\mathcal{Y}, \tag{48}$$

$$0 \le q_t \le 1, \ t = 1, \ldots, T, \tag{49}$$

$$\boldsymbol{f} \in \mathcal{F}, \tag{50}$$

where $q_{t,y_t}$ is a shortcut to $q_t[y_t = 1] + (1-q_t)[y_t = 0]$ and labels $\tilde{\boldsymbol{y}}$ and $\hat{\boldsymbol{y}}$ serve as the selected labels $i$ and $j$, respectively.

The calibration function $H_{\Phi_{\text{quad}}, L_{\text{Ham},T}, \mathcal{F}_{\text{Ham},T}}(\varepsilon) = \frac{\varepsilon^2}{8T}$ in the formulation of this proposition matches the lower bound provided by Theorem 7 in Section D.3. Thus, it suffices to construct a feasible w.r.t. (46)-(50) assignment of variables $\boldsymbol{f}$, $\boldsymbol{q}$ and labels $\tilde{\boldsymbol{y}}$, $\hat{\boldsymbol{y}}$ such that the objective equals the lower bound.

It suffices to simply set $\tilde{\boldsymbol{y}}$ to all zeros and $\hat{\boldsymbol{y}}$ to all ones. In this case, the constraints (46) and (47) take the simplified form:

$$\frac{1}{T}\sum_{t=1}^T (1-2q_t) \ge \varepsilon, \tag{51}$$

$$q_t \le \frac{1}{2}, \ t = 1, \ldots, T. \tag{52}$$

We now set $q_t := \frac{1}{2} - \frac{\varepsilon}{2}$, $t = 1, \dots, T$, and $\boldsymbol{f} := -\frac{1}{2}\mathbf{1}_k$. This point is clearly feasible when $0 \le \varepsilon \le 1$, so it remains to compute the value of the objective. We complete the proof by writing (let $w$ be the count of ones in an assignment $\boldsymbol{y}$):

$$\frac{1}{2k}\sum_{\boldsymbol{y}\in\mathcal{Y}}^{k}\left(f_{\boldsymbol{y}}+1-\frac{1}{T}\sum_{t=1}^{T}q_{t,y_t}\right)^2 =$$

$$\frac{1}{2k}\sum_{w=0}^{T}\binom{T}{w}\left(\frac{1}{2}-\frac{1}{T}(w(\frac{1}{2}-\frac{\varepsilon}{2})+(T-w)(\frac{1}{2}+\frac{\varepsilon}{2}))\right)^2 =$$

$$\frac{1}{2k}\sum_{w=0}^{T}\binom{T}{w}(\frac{\varepsilon}{2}-\frac{w\varepsilon}{T})^2 = \frac{\varepsilon^2}{2k}\sum_{w=0}^{T}\binom{T}{w}(\frac{1}{4}-\frac{w}{T}+\frac{w^2}{T^2}) =$$

$$\frac{\varepsilon^2}{2k}(\frac{1}{4}2^T - \frac{1}{T}T2^{T-1} + \frac{1}{T^2}T(T+1)2^{T-2}) = \frac{\varepsilon^2}{8T},$$

where we use the equality $k = 2^T$ and the identities $\sum_{t=0}^{T}\binom{T}{t} = 2^T$, $\sum_{t=0}^{T}t\binom{T}{t} = T2^{T-1}$, $\sum_{t=0}^{T}t^2\binom{T}{t} = T(T+1)2^{T-2}$. $\qquad\square$

### E.4 Mixed 0-1 and block 0-1 loss

Recall that $L_{01,b,\eta}$ is the convex combination of the 0-1 loss and the block 0-1 loss with $b$ blocks, i.e., $L_{01,b,\eta} = \eta L_{01} + (1-\eta)L_{01,b}$, $0 \le \eta \le 1$. Let all the blocks be of the same size $s = \frac{k}{b} \ge 2$. In this section, we compute the calibration functions for the case of unconstrained scores (Proposition 16) and for the case when scores belong to the column span of the loss matrix (Proposition 17).

**Proposition 16.** *If there are no constraints on scores $\boldsymbol{f}$ then the calibration function*

$$H_{\Phi_{\mathrm{quad}},L_{01,b,\eta},\mathbb{R}^k}(\varepsilon) = \begin{cases} \frac{\varepsilon^2}{4k}, & \varepsilon \le \frac{\eta}{1-\eta}, \\ \frac{\varepsilon^2 s}{2k(s+1)} - \frac{\eta(\varepsilon+1)(s-1)}{4k(s+1)}(2\varepsilon - \varepsilon\eta - \eta) & \frac{\eta}{1-\eta} \le \varepsilon \le 1 \end{cases}$$

*shows that the surrogate is consistent.*

Note that when $\eta = 0$, we have $H(\varepsilon) = \frac{\varepsilon^2}{4k}\frac{2s}{s+1}$ as in Proposition 13. When $\eta \ge 0.5$ we have $H(\varepsilon) = \frac{\varepsilon^2}{4k}$, which matches Proposition 12.

*Proof.* This proof is very similar to the proof of Proposition 13, but technically more involved.

We start by noting that the $i$-th element of the vector $(L_{01,b,\eta})\boldsymbol{q}$ equals

$$\sum_{j:\,b(j)\neq b(i)}(1-\eta)q_j + \sum_{j:\,j\neq i}\eta q_j = \eta(1-q_i) + (1-\eta)(1-Q_{b(i)}), \tag{53}$$

where for $b(i)$ and $Q_v$ we reuse the notation defined in the proof of Proposition 13. By combining this with Lemmas 9 and 10, we get

$$H_{ij}(\varepsilon)=\min_{\boldsymbol{f},\boldsymbol{q}}\frac{1}{2k}\sum_{v=1}^{b}\sum_{c\in\mathcal{Y}_v}(f_c + 1 - \eta q_c - (1-\eta)Q_{b(c)})^2, \tag{54}$$

$$\text{s.t. } \eta(q_i - q_j) + (1-\eta)(Q_{b(i)} - Q_{b(j)}) \ge \varepsilon,$$
$$\eta(q_i - q_c) + (1-\eta)(Q_{b(i)} - Q_{b(c)}) \ge 0, \forall c$$
$$f_j \ge f_c, \ \forall c,$$
$$\sum_{c=1}^{k}q_c = 1, \quad q_c \ge 0, \ \forall c.$$

The blocks are all of the same size so we need to consider just the two cases: 1) the selected labels belong to the same block, i.e., $b(i) = b(j)$; 2) the selected labels belong to the two different blocks, i.e., $b(i) \neq b(j)$.

The first case can be proven by a straight forward generalization of the proof of Proposition 12. Given that the loss value is bounded by 1, the maximal possible value of $\varepsilon$ when the constraints can be feasible equals $\eta$. Thus, we have $H_{ij}(\varepsilon) = \frac{\varepsilon^2}{4k}$ for $\varepsilon \leq \eta$ and $+\infty$ otherwise.

We will now proceed to the second case $b(i) \neq b(j)$. We show that

$$H_{ij}(\varepsilon) = \begin{cases} \frac{\varepsilon^2}{4k}, & \text{for } \varepsilon \leq \frac{\eta}{1-\eta}, \\ \frac{\varepsilon^2 s}{2k(s+1)} - \frac{\eta(\varepsilon+1)(s-1)}{4k(s+1)}(2\varepsilon - \varepsilon\eta - \eta), & \text{otherwise.} \end{cases}$$

Similarly to the arguments used in Propositions 12 and 13, we claim that there is an optimal point of (54) such that $q_c^* = 0$, $c \notin \{i,j\}$; $q_i^* = 0.5 + \frac{\varepsilon}{2}$; $q_j^* = 0.5 - \frac{\varepsilon}{2}$; and $f_c^* = -1$ for $c \notin \mathcal{Y}_{ij} := \mathcal{Y}_{b(i)} \cup \mathcal{Y}_{b(j)}$.

First, we will show that we can consider only configurations with all the probability mass on the two selected blocks $b(i)$ and $b(j)$. Given any optimal point $\boldsymbol{f}^*$ and $\boldsymbol{q}^*$, the operation (with $\delta = \sum_{c \notin \mathcal{Y}_{ij}} q_c^*$)

$$f_i^* := f_i^* + \frac{\delta}{2}, \qquad\qquad q_i^* := q_i^* + \frac{\delta}{2},$$
$$f_j^* := f_j^* + \frac{\delta}{2}, \qquad\qquad q_j^* := q_j^* + \frac{\delta}{2},$$
$$f_c^* := -1, c \notin \mathcal{Y}_{ij} \qquad\qquad q_c^* := 0, c \notin \mathcal{Y}_{ij}$$
$$f_c^* := f_c^* + (1-\eta)\frac{\delta}{2}, \ \ c \in \mathcal{Y}_{ij} \setminus \{i,j\}$$

can only decrease the objective of (54) because the summands corresponding to the unselected $b - 2$ blocks are set to zero. All the constraints stay feasible and the values corresponding to the blocks $b(i)$ and $b(j)$ do not change. The last operation is required, because the values $Q_{b(i)}, Q_{b(j)}$ change when we change $q_i$ and $q_j$. Adding $(1-\eta)\frac{\delta}{2}$ to some scores compensates this and cannot violate the constraints because $f_j^*$ goes up by $\frac{\delta}{2} \geq (1-\eta)\frac{\delta}{2}$.

Now we will show that it is possible to move all the mass to the two selected labels $i$ and $j$. We cannot simply move the mass within one block, but need to create some overflow and move it to another block in a specific way. Consider $\delta := q_a^*$, which is some non-zero mass on a non-selected label of the block $b(i)$. Then, the operation

$$f_i^* := f_i^* + \delta\frac{\eta}{2}, \qquad\qquad q_i^* := q_i^* + \delta(1 - \frac{\eta}{2}),$$
$$f_j^* := f_j^* + \delta\frac{\eta}{2}, \qquad\qquad q_j^* := q_j^* + \delta\frac{\eta}{2},$$
$$f_a^* := f_a^* + \delta\frac{\eta}{2}(\eta - 3), \qquad\qquad q_a^* := q_a^* - \delta = 0,$$
$$f_c^* := f_c^* - \delta\frac{\eta}{2}(1-\eta), \ \ c \in \mathcal{Y}_i \setminus \{i,a\}$$
$$f_c^* := f_c^* + \delta\frac{\eta}{2}(1-\eta), \ \ c \in \mathcal{Y}_j \setminus \{j\}$$

does no change the objective value of (54) because the quantities $f_c + 1 - \eta q_c - (1-\eta)Q_{b(c)}$, $c \in \mathcal{Y}_{ij}$, stay constant and all the constraints of (54) stay feasible. We repeat this operation for all $a \in \mathcal{Y}_{b(i)} \setminus \{i\}$ and, thus, move all the probability mass within the block $b(i)$ to the label $i$. In the block $b(j)$, an analogous operation can move all the mass to the label $j$.

It remains to show that $q_i^* - q_j^* = \varepsilon$. Indeed, if $q_i^* - q_j^* = \delta' > \varepsilon$, the operation analogous to (39)

$$f_i^* := f_i^* - \frac{\delta'-\varepsilon}{2}, \qquad\qquad q_i^* := q_i^* - \frac{\delta'-\varepsilon}{2},$$
$$f_j^* := f_j^* + \frac{\delta'-\varepsilon}{2}, \qquad\qquad q_j^* := q_j^* + \frac{\delta'-\varepsilon}{2},$$
$$f_c^* := f_c^* - (1-\eta)\frac{\delta'-\varepsilon}{2}, c \in \mathcal{Y}_{b(i)} \setminus \{i\},$$
$$f_c^* := f_c^* + (1-\eta)\frac{\delta'-\varepsilon}{2}, c \in \mathcal{Y}_{b(j)} \setminus \{j\}$$

can always set $q_i^* - q_j^* = \varepsilon$, and thus $q_i^* = 0.5 + \frac{\varepsilon}{2}$ and $q_j^* = 0.5 - \frac{\varepsilon}{2}$. After this operation, all the scores of the block $b(i)$ go down and all the scores of the block $b(j)$ go up at most as much as $f_j^*$, so the constraints $f_j \geq f_c$ cannot get violated.

We now proceed with the computation of $H_{ij}(\varepsilon)$. First, we note that convexity and symmetries of (54) implies that all the non-selected scores within each block are equal.[13] Denote the scores of the

non-selected labels of the block $b(i)$ by $f'_i$, and the scores of the non-selected labels of the block $b(j)$ by $f'_j$.

Analogous to all the previous propositions, the truncation argument gives us that all the values $f^*_c$ belong to the segment $[-1, -0.5 + \frac{\varepsilon}{2}]$. For all the optimal values $f^*_c$, $c \neq j$, the following identity holds:

$$f^*_c = \begin{cases} f^*_j, & \text{if } \eta q^*_c + (1-\eta)Q^*_{b(c)} - 1 \geq f^*_j, \\ \eta q^*_c + (1-\eta)Q^*_{b(c)} - 1, & \text{otherwise.} \end{cases}$$

Given that $f^*_i$ wants to equal the maximal possible value $-0.5 + \frac{\varepsilon}{2}$, it implies that $f^*_i = f^*_j$. Denote this value by $f$.

By, plugging the values of $\boldsymbol{q}^*$ and $\boldsymbol{f}^*$ provided above into the objective of (54), we get

$$\tfrac{1}{2k}\Big((f + 0.5 - \tfrac{\varepsilon}{2})^2 + (s-1)(f'_i + 1 - (1-\eta)(0.5 + \tfrac{\varepsilon}{2}))^2 +$$

$$(f + 0.5 + \tfrac{\varepsilon}{2})^2 + (s-1)(f'_j + 1 - (1-\eta)(0.5 - \tfrac{\varepsilon}{2}))^2\Big). \tag{55}$$

By minimizing (55) without constraints, we get $f^* = -0.5$, $f'^*_i = \frac{1}{2}(1 + \varepsilon)(1 - \eta) - 1$, $f'^*_j = \frac{1}{2}(1 - \varepsilon)(1 - \eta) - 1$. We now need to compare $f'^*_i$ and $f'^*_j$ with $f^*$ to satisfy the constraints $f^* \geq f'^*_i$ and $f^* \geq f'^*_j$. First, we have that

$$f^* - f'^*_j = \tfrac{1}{2}(\eta + \varepsilon - \eta\varepsilon) \geq 0, \text{ for } 0 \leq \varepsilon \leq 1 \text{ and } 0 \leq \eta \leq 1.$$

Second, we have

$$f^* - f'^*_i = \tfrac{1}{2}(\eta - \varepsilon + \eta\varepsilon) \geq 0, \text{ for } 0 \leq \varepsilon \leq \tfrac{\eta}{1-\eta} \text{ and } 0 \leq \eta \leq 1.$$

We can now conclude that when $\varepsilon \leq \frac{\eta}{1-\eta}$ we have both $f'_i$ and $f'_j$ equal to their unconstrained minimum points leading to $H_{ij}(\varepsilon) = \frac{\varepsilon^2}{4k}$.

Now, consider the case $\varepsilon > \frac{\eta}{1-\eta}$. We have the constraint $f \geq f'_i$ violated, so at the minimum we have $f'_i = f$. The new unconstrained minimum w.r.t. $f$ equals $f^* = \frac{1}{s+1}(-1 - (s-1)(1 - \frac{1}{2}(1-\eta)(1-\varepsilon)))$. We now show that the inequality $f^* \geq f'^*_j$ still holds. We have

$$f^* - f'^*_j = \tfrac{\eta + \varepsilon s - \eta\varepsilon s}{s+1} \geq 0, \text{ for } 0 \leq \varepsilon \leq 1 \text{ and } 0 \leq \eta \leq 1.$$

Substitution of $f'^*_i = f^*$ and $f'^*_j$ into (55) gives us

$$\tfrac{1}{k}\left(\tfrac{\varepsilon^2 s}{2(s+1)} - \tfrac{\eta(\varepsilon+1)(s-1)}{4(s+1)}(2\varepsilon - \varepsilon\eta - \eta)\right),$$

which equals $H_{ij}(\varepsilon)$ for $1 \geq \varepsilon > \frac{\eta}{1-\eta}$.

Comparing cases 1 and 2, we observe that $H_{ij}(\varepsilon)$ from case 2 is never larger than the one of case 1, thus case 2 provides the overall calibration function $H_{ij}(\varepsilon)$. $\qquad\square$

**Proposition 17.** *If the scores $\boldsymbol{f}$ are constrained to be equal inside the blocks, i.e. belong to the subspace $\mathcal{F}_{01,b} = \mathrm{span}(L_{01,b}) \subseteq \mathbb{R}^k$, then the calibration function*

$$H_{\Phi_{\text{quad}}, L_{01,b,\eta}, \mathcal{F}_{01,b}}(\varepsilon) = \begin{cases} \dfrac{(\varepsilon - \frac{\eta}{2})^2}{4b} \dfrac{(\frac{\eta b}{k} + 1 - \eta)^2}{(1 - \frac{\eta}{2})^2}, & \frac{\eta}{2} \leq \varepsilon \leq 1, \\ 0, & 0 \leq \varepsilon \leq \frac{\eta}{2} \end{cases}$$

*shows that the surrogate is consistent up to level $\frac{\eta}{2}$.*

When $\eta = 0$, we have $H(\varepsilon) = \frac{\varepsilon^2}{4b}$ as in Proposition 14. When $\eta > 0$ we have $H(\varepsilon) = 0$ for small $\varepsilon$, which corresponds to the case of inconsistent surrogate (0-1 loss and constrained scores).

*Proof.* This proof combines ideas from Proposition 16 and Proposition 14.

Note that contrary to all the previous results, Lemma 9 is not applicable, because, for $b < k$, we have that $\mathrm{span}(L_{01,b,\eta}) = \mathbb{R}^k \not\subset \mathcal{F}_{01,b} = \mathrm{span}(L_{01,b})$.

We now derive an analog of Lemma 9 for this specific case. We define the subspace of scores $\mathcal{F}_{01,b} = \{F\boldsymbol{\theta} \mid \boldsymbol{\theta} \in \mathbb{R}^b\}$ with a matrix $F := F_{01,b} \in \mathbb{R}^{k \times b}$ with columns containing the indicator vectors of the blocks. We have $F^{\mathsf{T}}F = s\mathbf{I}_b$ and thus $(F^{\mathsf{T}}F)^{-1} = \frac{1}{s}\mathbf{I}_b$. We shortcut the loss matrix $L_{01,b,\eta}$ to $L$ and rewrite it as

$$L = \eta L_{01} + (1-\eta)L_{01,b} = \mathbf{1}_k\mathbf{1}_k^{\mathsf{T}} - \eta\mathbf{I}_k - (1-\eta)FF^{\mathsf{T}}.$$

By redoing the derivation of Lemma 9, we arrive at a different excess surrogate:

$$\phi(\boldsymbol{f}(\boldsymbol{\theta}),\boldsymbol{q}) = \frac{1}{2k}(s\boldsymbol{\theta}^{\mathsf{T}}\boldsymbol{\theta} + 2\boldsymbol{\theta}^{\mathsf{T}}F^{\mathsf{T}}L\boldsymbol{q}) + r(\boldsymbol{q}),$$

$$\boldsymbol{\theta}^* := \operatorname{argmin}_{\boldsymbol{\theta}} \phi(\boldsymbol{f}(\boldsymbol{\theta}),\boldsymbol{q}) = -\frac{1}{s}F^{\mathsf{T}}L\boldsymbol{q},$$

$$\delta\phi(\boldsymbol{f}(\boldsymbol{\theta}),\boldsymbol{q}) = \frac{1}{2k}(s\boldsymbol{\theta}^{\mathsf{T}}\boldsymbol{\theta} + 2\boldsymbol{\theta}^{\mathsf{T}}F^{\mathsf{T}}L\boldsymbol{q} + \frac{1}{s}\boldsymbol{q}^{\mathsf{T}}L^{\mathsf{T}}FF^{\mathsf{T}}L\boldsymbol{q})$$

$$= \frac{s}{2k}\|\boldsymbol{\theta} + \frac{1}{s}F^{\mathsf{T}}L\boldsymbol{q}\|_2^2$$

$$= \frac{s}{2k}\sum_{v=1}^{s}(\theta_v + 1 - (1-\eta)Q_v - \frac{\eta}{s}Q_v)^2,$$

where $Q_v = \sum_{c \in \mathcal{Y}_v} q_c$ is the total probability mass on block $v$ and $\mathcal{Y}_v \subset \mathcal{Y}$ denotes the set of labels of block $v$.

Analogously to Proposition 16 we can now apply Lemma 10 and obtain $H_{ij}(\varepsilon)$.

$$H_{ij}(\varepsilon) = \min_{\boldsymbol{\theta},\boldsymbol{q}} \frac{s}{2k}\sum_{v=1}^{b}(\theta_v + 1 - (1-\eta)Q_v - \frac{\eta}{s}Q_v)^2, \tag{56}$$

$$\text{s.t. } \eta(q_i - q_j) + (1-\eta)(Q_{b(i)} - Q_{b(j)}) \geq \varepsilon,$$

$$\eta(q_i - q_c) + (1-\eta)(Q_{b(i)} - Q_{b(c)}) \geq 0, \forall c$$

$$\theta_{b(j)} \geq \theta_u, \ \forall u = 1, \dots, b,$$

$$\sum_{c=1}^{k} q_c = 1, \quad q_c \geq 0, \ \forall c.$$

The main difference to (54) consists in the fact that we now minimize w.r.t. $\boldsymbol{\theta}$ instead of $\boldsymbol{f}$.

Note that because of the way the predictor $\operatorname{pred}(\boldsymbol{f}(\boldsymbol{\theta}))$ resolves ties (among the labels with maximal scores it always picks the label with the smallest index), not all labels can be predicted. Specifically, only one label from each block can be picked. This argument allows us to assume that $b(i) \neq b(j)$ in the remainder of this proof.

First, let us prove the case for $\varepsilon \leq \frac{\eta}{2}$. We explicitly provide a feasible assignment of variables where the objective equals zero. We set $q_i = \frac{1}{2}$ and $q_c = \frac{1}{2(s-1)}$, $c \in \mathcal{Y}_{b(j)} \setminus \{j\}$. All the other labels (including $j$ and the unselected labels of the block $b(i)$) receive zero probability mass. This assignment of $\boldsymbol{q}$ implies $Q_{b(i)} = Q_{b(j)} = \frac{1}{2}$ and the zero mass on the other blocks. We also set $\theta_{b(i)}$ and $\theta_{b(j)}$ to $(1-\eta)\frac{1}{2} + \frac{\eta}{s}\frac{1}{2} - 1$ to ensure zero objective value. Verifying other feasibility constraints we have $\eta(q_i - q_j) + (1-\eta)(Q_{b(i)} - Q_{b(j)}) = \frac{\eta}{2} \geq \varepsilon$ and $\eta(q_i - q_c) + (1-\eta)(Q_{b(i)} - Q_{b(c)}) = \eta(\frac{1}{2} - \frac{1}{2(s-1)}) \geq 0$, $c \in \mathcal{Y}_{b(j)} \setminus \{j\}$. Other constraints are trivially satisfied.

Now, consider the case of $\varepsilon > \frac{\eta}{2}$. As usual, we claim the following values of the variables $\boldsymbol{f}$ and $\boldsymbol{q}$ result in an optimal point. We have $q_c^* = 0$, $c \notin \mathcal{Y}_{ij}$; $\theta_v^* = -1$, $v \notin \{b(i), b(j)\}$; and $q_i^* = Q_{b(i)}^* = \frac{1+\varepsilon-\eta}{2-\eta}$; $q_c^* = 0$, $c \in \mathcal{Y}_{b(i)} \setminus \{i\}$ (other labels in the block $b(i)$); $q_j^* = 0$, $q_c^* = \frac{1-\varepsilon}{(2-\eta)(s-1)}$, $c \in \mathcal{Y}_{b(j)} \setminus \{j\}$ (other labels in the block $b(j)$).

First, we will show that we can consider only configurations with all the probability mass on the two selected blocks $b(i)$ and $b(j)$. Given some optimal variables $\boldsymbol{f}^*$ and $\boldsymbol{q}^*$, the operation (with $\delta = \sum_{c \in \mathcal{Y} \setminus \mathcal{Y}_{ij}} q_c^*$)

$$q_c^* := 0, \ c \in \mathcal{Y} \setminus \mathcal{Y}_{ij}, \qquad q_i^* := q_i^* + \frac{\delta}{2}, \qquad q_j^* := q_j^* + \frac{\delta}{2},$$

$$\theta_v^* := -1, \ v \notin \{b(i), b(j)\},$$

$$\theta_{b(i)}^* := \theta_{b(i)}^* + \frac{\delta}{2}(1 - \eta + \frac{\eta}{s}),$$

$$\theta_{b(j)}^* := \theta_{b(j)}^* + \frac{\delta}{2}(1 - \eta + \frac{\eta}{s})$$

can only decrease the objective of (56) because the summands corresponding to the unselected $b - 2$ blocks are set to zero. All the constraints stay feasible and the values corresponding to the blocks $b(i)$ and $b(j)$ do not change.

Now, we move the mass within the two selected blocks. To start with, moving the mass within one block does not change the objective, because it depends only on $Q_{b(c)}$ and not on $\boldsymbol{q}$ directly. In the block $b(i)$, it is safe to increase $q_i$ and decrease the mass on the other labels, because $q_i$ enters the constraints with the positive sign and while the others enter with the negative sign. So we let $q_c = 0$ for $c \in \mathcal{Y}_{b(i)}/\{i\}$ and $Q_{b(i)} = q_i$. We also have $Q_{b(j)} = 1 - q_i$ as the mass on all other blocks is zero.

Moving mass within the block $b(j)$ is more complicated, as moving mass to some label $c$ of this block might violate the constraints of (56) on $q_i$. We start by considering the first constraint in (56), using $Q_{b(j)} = 1 - q_i$, we get:

$$q_i \geq \varepsilon + \eta q_j + (1 - \eta)(1 - q_i). \tag{57}$$

By using $q_j \geq 0$ and $\varepsilon \geq \frac{\eta}{2}$, the inequality (57) implies that $q_i \geq \frac{1}{2}$ and thus that

$$q_c \leq Q_{b(j)} \leq \tfrac{1}{2} \quad \forall c \in \mathcal{Y}_{b(j)}. \tag{58}$$

Now the second constraint of (56) that we want to satisfy is:

$$q_i \geq \eta q_c + (1 - \eta)Q_{b(j)} \quad \forall c \in \mathcal{Y}_{b(j)}. \tag{59}$$

Using (58), we have that the RHS of (59) is $\leq 1/2$, and so since $q_i \geq 1/2$, we have that (59) is satisfied for any valid mass distribution on block $b(j)$ (i.e. such that $Q_{b(j)} \leq 1/2$). Using $q_j = 0$ gives the most possibilities for the value of $q_i$ in the constraint (57). Moreover, the constraint (57) is more stringent than the constraint (59), i.e. if it is satisfied, the second one is also satisfied; so we focus only on the first constraint.

As in the proof of all other propositions, we can make the constraint (57) an equality for the optimum by generalizing the transformation of (39) which makes the constraint tight without changing the objective and maintaining feasibility. So (57) as an equality with $q_j = 0$ yields the value

$$q_i^* = \tfrac{1 + \varepsilon - \eta}{2 - \eta}.$$

So to summarize at this point, we have $q_j^* = 0$; $q_c^* = 0, c \in \mathcal{Y}_{b(i)} \setminus \{i\}$; $q_c^* = 0, \mathcal{Y}_{b(i)} \notin \{b(i), b(j)\}$. $q_i^* = \frac{1 + \varepsilon - \eta}{2 - \eta}$ and $Q_{b(j)} = 1 - q_i^*$. The precise distribution of mass for $c \in \mathcal{Y}_{b(j)}/\{j\}$ does not matter (any distribution is feasible and does not influence the objective, only the total mass matters), but for concreteness, we can choose them to all have the same mass yielding $q_c^* = \frac{1 - \varepsilon}{(2 - \eta)(s - 1)}$, $c \in \mathcal{Y}_{b(j)} \setminus \{j\}$.

We now finish the computation of $H_{ij}(\varepsilon)$. First, we note that, due to the truncation argument similar to the one mentioned in the paragraph after (39), we have both $\theta_i^*$ and $\theta_j^*$ in the segment $[(1 - \eta)Q_{b(j)}^* + \frac{\eta}{s}Q_{b(j)}^* - 1, (1 - \eta)Q_{b(i)}^* + \frac{\eta}{s}Q_{b(i)}^* - 1]$ and since $\theta_j^* \geq \theta_i^*$, we have $\theta_j^* = \theta_i^* =: \theta$ at the optimum.

Substituting the values $Q_{b(i)}^*$ and $Q_{b(j)}^*$ provided above into the objective of (56) and performing unconstrained minimization w.r.t. $\theta$ (we use the help of MATLAB symbolic toolbox to set the derivative to zero) we get

$$\theta^* = -\tfrac{s - \eta + \eta s}{2s}$$

and, consequently,

$$H_{ij}(\varepsilon) = \frac{s(\varepsilon - \frac{\eta}{2})^2(\frac{\eta}{s} + 1 - \eta)^2}{4k(1 - \frac{\eta}{2})^2},$$

which finishes the proof. $\qquad\square$

## F Constants in the SGD rate

To formalize the learning difficulty by bounding the required number of iterations to get a good value of the risk (Theorem 6), we need to bound the constants $D$ and $M$. In this section, we provide a

way to bound these constants for the quadratic surrogate $\Phi_{\text{quad}}$ (12) under a simplifying assumption slightly stronger than the well-specified model Assumption 4.

Consider the family of score functions $\mathfrak{F}_{F,\mathcal{H}}$ defined via an explicit feature map $\psi(\boldsymbol{x}) \in \mathcal{H}$, i.e., $\mathfrak{f}_W(\boldsymbol{x}) = FW\psi(\boldsymbol{x})$, where a matrix $F \in \mathbb{R}^{k \times r}$ defines the structure and an operator (which we think of as a matrix with one dimension being infinite) $W : \mathcal{H} \to \mathbb{R}^r$ contains the learnable parameters. Then the surrogate risk can be written as

$$\mathcal{R}_\Phi(\mathfrak{f}_W) = \mathbf{E}_{(\boldsymbol{x},\boldsymbol{y})\sim\mathcal{D}} \tfrac{1}{2k} \|FW\psi(\boldsymbol{x}) + L(:,\boldsymbol{y})\|_{\mathbb{R}^k}^2$$

and its stochastic w.r.t. $(\boldsymbol{x},\boldsymbol{y})$ gradient as

$$\boldsymbol{g}_{\boldsymbol{x},\boldsymbol{y}}(W) = \tfrac{1}{k} F^\mathsf{T}(FW\psi(\boldsymbol{x}) + L(:,\boldsymbol{y}))\psi(\boldsymbol{x})^\mathsf{T} \qquad (60)$$

where $L(:,\boldsymbol{y})$ denotes the column of the loss matrix corresponding to the correct label $\boldsymbol{y}$. Note that computing the stochastic gradient requires performing products $F^\mathsf{T}F$ and $F^\mathsf{T}L(:,\boldsymbol{y})$ for which direct computation is intractable when $k$ is exponential, but which can be done in closed form for the structured losses we consider (the Hamming and block 0-1 loss). More generally, these operations require suitable inference algorithms.

To derive the constants, we use a simplifying assumption stronger than Assumption 4 in the case of quadratic surrogate: we assume that the conditional $q_c(\boldsymbol{x})$, seen as a function of $\boldsymbol{x}$, belongs to the RKHS $\mathcal{H}$, which by the reproducing property implies that for each $c = 1, \ldots, k$, there exists $v_c \in \mathcal{H}$ such that $q_c(\boldsymbol{x}) = \langle v_c, \psi(\boldsymbol{x})\rangle_\mathcal{H}$ for all $\boldsymbol{x} \in \mathcal{X}$. Concatenating all $v_c$, we get an operator $V : \mathcal{H} \to \mathbb{R}^k$. To derive the bound, we also assume that $\sum_{c=1}^k \|v_c\|_\mathcal{H} \le Q_{\max}$ and $\|\psi(\boldsymbol{x})\|_\mathcal{H} \le R$ for all $\boldsymbol{x} \in \mathcal{X}$. In the following, we use the notation $\boldsymbol{q}_{\boldsymbol{x}}$ to denote the vector in $\mathbb{R}^k$ with components $q_c(\boldsymbol{x})$, $c = 1, \ldots, k$, for a fixed $\boldsymbol{x}$, and thus $\boldsymbol{q}_{\boldsymbol{x}} = V\psi(\boldsymbol{x})$.

Under these assumptions, we can write the theoretical minimum of the surrogate risk. The gradient of the surrogate risk gives

$$k\nabla_W \mathcal{R}_\Phi(\mathfrak{f}_W) = F^\mathsf{T}FW\mathbf{E}_{\boldsymbol{x}\sim\mathcal{D}_\mathcal{X}}(\psi(\boldsymbol{x})\psi(\boldsymbol{x})^\mathsf{T}) + F^\mathsf{T}L\mathbf{E}_{\boldsymbol{x}\sim\mathcal{D}_\mathcal{X}}(\boldsymbol{q}_{\boldsymbol{x}}\psi(\boldsymbol{x})^\mathsf{T})$$
$$= F^\mathsf{T}FW\mathbf{E}_{\boldsymbol{x}\sim\mathcal{D}_\mathcal{X}}(\psi(\boldsymbol{x})\psi(\boldsymbol{x})^\mathsf{T}) + F^\mathsf{T}LV\mathbf{E}_{\boldsymbol{x}\sim\mathcal{D}_\mathcal{X}}(\psi(\boldsymbol{x})\psi(\boldsymbol{x})^\mathsf{T})$$
$$= (F^\mathsf{T}FW + F^\mathsf{T}LV)\,\mathbf{E}_{\boldsymbol{x}\sim\mathcal{D}_\mathcal{X}}(\psi(\boldsymbol{x})\psi(\boldsymbol{x})^\mathsf{T}).$$

Setting the content of the parenthesis to zero gives that $W^* = -(F^\mathsf{T}F)^\dagger F^\mathsf{T}LV$ is a solution to the stationary condition equation $\nabla_W \mathcal{R}_\Phi(\mathfrak{f}_W) = 0$.

We can now bound the Hilbert-Schmidt norm of this choice of optimal parameters $W^*$ as

$$\|W^*\|_{HS} = \|(F^\mathsf{T}F)^\dagger F^\mathsf{T}LV\|_{HS}$$
$$\le \|(F^\mathsf{T}F)^\dagger F^\mathsf{T}\|_{HS}\|LV\|_{HS} \qquad \text{//submultiplicativity of } \|\cdot\|_{HS}$$
$$\le \sqrt{r}\|(F^\mathsf{T}F)^\dagger F^\mathsf{T}\|_2\|LV\|_{HS} \qquad \text{//connection of } \|\cdot\|_{HS} \text{ and } \|\cdot\|_2 \text{ via } r = \text{rank}(F)$$
$$= \sqrt{r}\sigma_{\min}^{-1}(F)\|LV\|_{HS} \qquad \text{//rotation invariance of } \|\cdot\|_2$$
$$\le \sqrt{r}\sigma_{\min}^{-1}(F)\sqrt{k}L_{\max}Q_{\max} =: D \qquad \text{//the definition of } \|\cdot\|_{HS} \text{ and triangular inequality}$$

where $\|\cdot\|_{HS}$ and $\|\cdot\|_2$ denote the Hilbert-Schmidt and spectral norms, respectively, and $\sigma_{\min}^{-1}(F)$ stands for the smallest singular value of the matrix $F$. The last inequality follows from the definition of the Hilbert-Schmidt norm $\|LV\|_{HS}^2 = \sum_{i=1}^k \|\sum_{c=1}^k L(i,c)v_c\|_\mathcal{H}^2$ and from the triangular inequality $\|\sum_{c=1}^k L(i,c)v_c\|_\mathcal{H} \le \sum_{c=1}^k |L(i,c)|\|v_c\|_\mathcal{H} \le L_{\max}Q_{\max}$ thus giving $\|LV\|_{HS} \le \sqrt{k}L_{\max}Q_{\max}$.

Analogously, we now bound the Hilbert-Schmidt norm of the stochastic gradient $\boldsymbol{g}_{\boldsymbol{x},\boldsymbol{y}}(W)$.

$$\|\boldsymbol{g}_{\boldsymbol{x},\boldsymbol{y}}(W)\|_{HS} \le \tfrac{1}{k}\|F^\mathsf{T}FW\psi(\boldsymbol{x}) + F^\mathsf{T}L(:,\boldsymbol{y}))\|_2\|\psi(\boldsymbol{x})\|_\mathcal{H}$$
$$\le \tfrac{1}{k}(\|F^\mathsf{T}FW\psi(\boldsymbol{x})\|_2 + \|F^\mathsf{T}L(:,\boldsymbol{y}))\|_2)\|\psi(\boldsymbol{x})\|_\mathcal{H}$$
$$\le \tfrac{1}{k}(\|F^\mathsf{T}F\|_2\|W\|_{HS}\|\psi(\boldsymbol{x})\|_\mathcal{H} + \|F\|_2\|L(:,\boldsymbol{y}))\|_2)\|\psi(\boldsymbol{x})\|_\mathcal{H}$$
$$\le \tfrac{1}{k}\sigma_{\max}^2(F)DR^2 + \tfrac{1}{k}\sigma_{\max}(F)\sqrt{k}L_{\max}R =: M$$

where $R$ is an upper bound on $\|\psi(\boldsymbol{x})\|_\mathcal{H}$ and $\sigma_{\max}(F)$ is a maximal singular value of $F$. Here the first inequality follows from the fact that the rank of $\boldsymbol{g}_{\boldsymbol{x},\boldsymbol{y}}(W)$ equals 1 and from submultiplicativity of

the spectral norm. We also use the inequality $\|W\psi(\boldsymbol{x})\|_2 \le \|W\|_{HS}\|\psi(\boldsymbol{x})\|_{\mathcal{H}}$, which follows from the properties of the Hilbert-Schmidt norm.

The bound of Theorem 5 contains the quantity $DM$ and the step size of ASGD depends on $\frac{D}{M}$, so, to be practical, both quantities cannot be exponential (for numerical stability; but the important quantity is the number of iterations from Theorem 6). We have

$$DM = \kappa^2(F)R^2 r L_{\max}^2 Q_{\max}^2 + \kappa(F)R\sqrt{r}L_{\max}^2 Q_{\max} = L_{\max}^2 \xi(\kappa(F)\sqrt{r}RQ_{\max}), \quad \xi(z) = z^2 + z,$$

$$\frac{M}{D} = \frac{\sigma_{\max}^2(F)}{k}R^2 + \frac{\sigma_{\max}(F)\sigma_{\min}(F)}{k}\frac{R}{Q_{\max}\sqrt{r}}$$

where $\kappa(F) = \frac{\sigma_{\max}}{\sigma_{\min}}$ is the condition number of $F$. Note that the quantity $DM$ is invariant to the scaling of the matrix $F$. The quantity $\frac{D}{M}$ scales proportionally to the square of the scale of $F$ and thus rescaling $F$ can always bring it to $\tilde{O}(1)$. For the rest of the analysis, we consider $R$ and $Q_{\max}$ to be well-behaved constants and thus focus on the dependence of the quantity $DM$ on $F$ and $L$.

### F.1 Constants for specific losses

We now estimate the product $DM$ from (18) for the 0-1, block 0-1 and Hamming losses. For the definition of the losses and the corresponding matrices $F$, we refer to Section D.3.

**0-1 loss.** For the 0-1 loss $L_{01}$ and $F = \mathbf{I}_k$, we have $L_{\max} = 1$, $r = k$, $\sigma_{\min} = \sigma_{\max} = 1$, thus $DM = O(k)$ is very large leading to very slow convergence of ASGD.

**Block 0-1 loss.** For the block 0-1 loss $L_{01,b}$ and matrix $F_{01,b}$, we have $L_{\max} = 1$, $r = b$, $\sigma_{\min} = \sigma_{\max} = \sqrt{s}$, thus $DM = O(b)$.

**Hamming loss.** For the Hamming loss, we have $L_{\max} = 1$, $r = \log_2 k + 1$, $\kappa(F_{\mathrm{Ham},T}) \le \log_2 k + 2$ (see the derivation in Section G). Finally, we have $DM = O(\log_2^3 k)$.

## G Properties of the basis of the Hamming loss

As defined in (35), the matrix $L_{\mathrm{Ham},T} \in \mathbb{R}^{k \times k}$ is the matrix of the Hamming loss between tuples of $T$ binary variables, and the number of labels equals $k = 2^T$. Also recall that $F_{\mathrm{Ham},T} := [\frac{1}{2}\mathbf{1}_{2^T}, \boldsymbol{h}^{(1)}, \ldots, \boldsymbol{h}^{(T)}]$, $(\boldsymbol{h}^{(t)})_{\hat{\boldsymbol{y}}} := [\hat{y}_t = 1]$, $t = 1, \ldots, T$. We have $\mathcal{F}_{\mathrm{Ham},T} = \mathrm{span}(F_{\mathrm{Ham},T}) = \mathrm{span}(L_{\mathrm{Ham},T})$ and $\mathrm{rank}(L_{\mathrm{Ham},T}) = \mathrm{rank}(F_{\mathrm{Ham},T}) = T+1$.

We now explicitly compute $\max_{i \ne j} \|P_{\mathcal{F}_{\mathrm{Ham},T}}\Delta_{ij}\|_2^2$. We shortcut $F_{\mathrm{Ham},T}$ by $F$ and compute

$$F^\mathsf{T}F = 2^{T-2}\begin{bmatrix} 1 & 1 & \cdots & 1 \\ 1 & 2 & 1 & \cdots \\ 1 & 1 & 2 & \cdots \\ \cdots & \cdots & \cdots & 1 \\ 1 & \cdots & 1 & 2 \end{bmatrix}. \tag{61}$$

We can compute the inverse matrix explicitly as well:

$$(F^\mathsf{T}F)^{-1} = 2^{2-T}\begin{bmatrix} 1+T & -1 & \cdots & -1 \\ -1 & 1 & 0 & \cdots \\ -1 & 0 & 1 & \cdots \\ \cdots & \cdots & \cdots & 0 \\ -1 & \cdots & 0 & 1 \end{bmatrix}. \tag{62}$$

The vector $F^\mathsf{T}\Delta_{ij}$ equals the difference of the two rows of $F$, i.e., $[0, c_1, \ldots, c_T]^\mathsf{T} \in \mathbb{R}^{T+1}$ with each $c_t \in \{-1, 0, +1\}$. We explicitly compute the square norm $\|P_{\mathcal{F}_{\mathrm{Ham},T}}\Delta_{ij}\|_2^2$:

$$\|P_{\mathcal{F}_{\mathrm{Ham},T}}\Delta_{ij}\|_2^2 = \Delta_{ij}^\mathsf{T}F(F^\mathsf{T}F)^{-1}F^\mathsf{T}\Delta_{ij} = [0, c_1, \ldots, c_T](F^\mathsf{T}F)^{-1}[0, c_1, \ldots, c_T]^\mathsf{T} = 2^{2-T}\sum_{t=1}^{T}c_t^2,$$

where the last equality follows from the identity submatrix of (62) and from the zero in the first position of the vector $F^\mathsf{T}\Delta_{ij}$. The quantity $\|P_{\mathcal{F}_{\mathrm{Ham},T}}\Delta_{ij}\|_2^2$ is maximized when none of $c_t$ equals zero, which is achievable, e.g., when the label $i$ corresponds to all zeros and the label $j$ to all ones. We now have $\max_{i \ne j} \|P_{\mathcal{F}_{\mathrm{Ham},T}}\Delta_{ij}\|_2^2 = \frac{4T}{2^T}$.

We now compute the smallest and largest eigenvalues of the Gram matrix (61) for $F_{\text{Ham},T}$. Ignoring the scaling factor $2^{T-2}$, we see by Gaussian elimination that the determinant and thus the product of all eigenvalues equals 1. If we subtract $\mathbf{I}_{T+1}$ the matrix becomes of rank 2, meaning that $T-1$ eigenvalues equal 1. The trace, i.e., the sum of the eigenvalues of (61), without the scaling factor $2^{T-2}$ equals $2T+1$. Summing up, we have $\lambda_{\min}\lambda_{\max}=1$ and $\lambda_{\min}+\lambda_{\max}=T+2$. We can now compute $\lambda_{\min}=\frac{1}{2}(T+2-\sqrt{T^2+4T})\in[\frac{1}{T+2},\frac{1}{T}]$ and $\lambda_{\max}=\frac{1}{2}(T+2+\sqrt{T^2+4T})\in[T+1,T+2]$. By putting back the multiplicative factor, we get $\sigma_{\min}=\sqrt{\lambda_{\min}}\geq\frac{\sqrt{k}}{2\sqrt{\log_2 k+2}}$ and $\sigma_{\max}=\sqrt{\lambda_{\max}}\leq\frac{\sqrt{k}}{2}\sqrt{\log_2 k+2}$, and thus the condition number is $\kappa\leq\log_2 k+2$.