[Reviews · NeurIPS 2017]

Reviewer 1



This paper presents a general theoretical framework on structure prediction via efficient convex Surrogate loss minimization. Allowing for the exponential number of classes, an interesting and specific result in Theorem 6 shows that the learning complexity is strongly dependent on the calibration function, and tells us that 0-1 loss for classification problem is ill-suited for the structured prediction with very large cardinality of labels. In summary, this paper provides sound theoretical results in terms of both statistical consistency and algorithmic convergence, and thus provides a theoretical insight on the design of algorithms for structure prediction via an efficient convex relaxation. Here I have some specific questions as follows: (1) Since the consistency of convex surrogates is well understood in the context of binary classification, what is the obvious difference or difficulty in term of theoretical proof between structure prediction with the exponential number of classes and the classical binary case? (2) In Section 4, this paper only considers the quadratic surrogate as special cases, it will be better that some robust loss functions such as the Huber loss and the pinball loss are also studied.

Reviewer 2



The paper studies the consistency of surrogate loss functions in structured prediction. Generally, minimizing the empirical risk directly in structured prediction is intractable, so instead people use surrogate losses like the structured hinge loss and the log loss. This paper is significant because it is the first to study under which circumstances minimizing these surrogate losses in the limit of infinite data implies minimizing the true task loss. The paper approaches the problem via calibration functions, functions that measures the smallest excess of the surrogate loss when the excess of the actual loss is larger than a given nonnegative value. The authors show how to analyze the calibration function for various surrogate and task loss function pairs in order to determine whether they are consistent. Perhaps the most interesting aspect of this analysis is that it formalizes common intuitions about different task losses for structured prediction, such as that learning with the 0-1 loss is hard. The paper and its supplementary material cover a lot of ground. A challenge for presentation is that parts of the paper read like a summary of the supplementary material with pointers to the appropriate sections.

Reviewer 3



The paper examines consistency of surrogate losses for multiclass prediction. The authors present their results using the formalism of structured prediction. Alas, there is no direct connection or exploitation of the separability of structured prediction losses. The paper is overly notated and variables are frequently overloaded. I got the feeling that the authors are trying to look mathematically fancy at the expense of readability. As for the technical content, I find the contribution in terms of new proof techniques quite marginal. The authors build on results by Toingo(Tong + Ingo) and the theorems seem lackluster to me. From algorithmic prospective the paper has rather little to offer. The quadratic loss is either intractable to calculate or in the case of separable it amounts to sum of squares over the vector L_j. There's no attempt to experimentally validate the calibrated losses. Also absent is a sincere discussion on the importance of calibration given the empirical success of overly-specified models. In "Contributions" paragraph the word "exponential" repeats over and over. It would not be clear to a reader who is not familiar with the setting what is the notion of exponential dependency. The sentence "... sets of possible predictions and correct outputs always coincide and do not depend on x." left me baffled. There is excessive overload of notation for L which is used as a function, specific losses, table, etc... minimal -> smallest / lowest We denote the y-th component of f with f y "... for now, we assume that a family of score functions F consists of all vector-valued Borel measurable functions f : X → F where F ⊆ R k ..." and then what ? Is it feasible to find inf φ ? "... equals the smallest excess of the conditional surrogate risk when the excess of the conditional actual risk is larger than ..." left me bewildered again. I feel that paper would have rather little appeal at NIPS and a natural venue would be COLT or ALT.v